# Maximizing Asynchronicity in Event-based Neural Networks

**Haiqing Hao[1], Nikola Zubić[2], Weihua He[1], Zhipeng Sui[1], Davide Scaramuzza[2], Wenhui Wang[*,1]**

[1]State Key Laboratory of Precision Measurement Technology and Instrument, Department of Precision Instrument, Tsinghua University [2]Robotics and Perception Group, University of Zurich
[*]Corresponding author
`wwh@tsinghua.edu.cn`

## Abstract

Event cameras deliver visual data with high temporal resolution, low latency, and minimal redundancy, yet their asynchronous, sparse sequential nature challenges standard tensor-based machine learning (ML). While the recent *asynchronous-to-synchronous* (A2S) paradigm aims to bridge this gap by asynchronously encoding events into learned features for ML pipelines, existing A2S approaches often sacrifice expressivity and generalizability compared to dense, synchronous methods. This paper introduces **EVA** (**EV**ent **A**synchronous feature learning), a novel A2S framework to generate highly expressive and generalizable event-by-event features. Inspired by the analogy between events and language, EVA uniquely adapts advances from language modeling in linear attention and self-supervised learning for its construction. In demonstration, EVA outperforms prior A2S methods on recognition tasks (DVS128-Gesture and N-Cars), and represents the first A2S framework to successfully master demanding detection tasks, achieving a 0.477 mAP on the Gen1 dataset. These results underscore EVA's potential for advancing real-time event-based vision applications.

## 1 Introduction

Event cameras have advantages in high temporal resolution (up to 1 $\mu$s) and low spatial redundancy (Gallego et al., 2020; Gehrig & Scaramuzza, 2024; He et al., 2024a;b), but challenge standard machine learning (ML) algorithms, which typically demand tensor-like inputs because of the asynchronous and sparse nature of the data (Sironi et al., 2018; Lagorce et al., 2016; Gehrig et al., 2019; Innocenti et al., 2021; Zubić et al., 2023). To bridge the gap between asynchronous data and synchronous ML algorithms, the *asynchronous-to-synchronous* (A2S) framework (Martin-Turrero et al., 2024) has recently emerged. By designing an efficient asynchronous encoder that recurrently integrates events into tensor-like features on an event-by-event basis and an on-demand sample strategy that feeds features to downstream ML algorithms at flexible rates, this paradigm successfully leverages the fine-grained temporal information and inherent sparsity of events while benefiting from advances in standard ML.

However, existing A2S work is limited in model expressivity and feature generalizability. On one hand, the asynchronous encoder relies on preliminary models as a compromise for computational efficiency considerations, which demands parallel training and equivalent recurrent inference. Therefore, the A2S method achieves suboptimal performance in complex tasks compared to dense approaches (Gehrig & Scaramuzza, 2023; Gehrig et al., 2019; Peng et al., 2023b; Fan et al., 2025) with event images. On the other hand, the features are end-to-end learned in a supervised manner and thus are task-specific. This limits their generalizability across diverse downstream applications.

To design an A2S framework with more expressivity and generalizability, we take inspiration from the analogy between events and language (Soydan et al., 2024) and leverage recent advances in natural language processing (NLP). First, events and language are both organized in sequence. Secondly, events record incremental visual changes, similar to words in a sentence that incrementally contribute to the semantic meaning. These two similarities inspire us to treat each event as one token and leverage recent linear attention (LA) architectures for efficient sequence modeling (Katharopou-

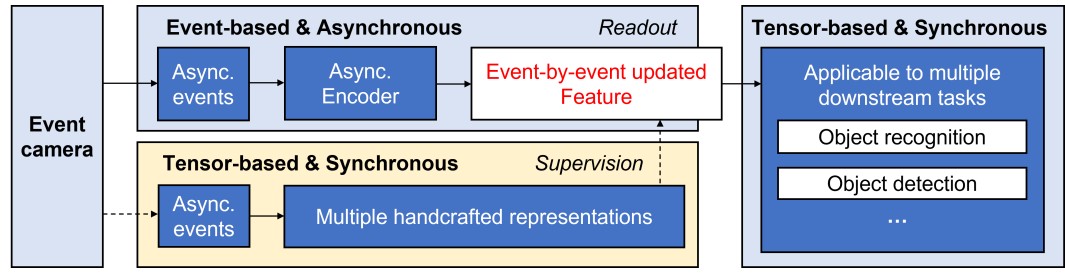

Figure 1: Overview of the proposed EVA A2S framework. It has an asynchronous LA-based encoder (top middle) producing event-by-event updated features for synchronous downstream tasks (right). These features are learned by self-supervision (bottom middle) and are event-by-event updated during inference to be sampled on demand by downstream tasks.

los et al., 2020; Yang et al., 2023a; Peng et al., 2023a; Sun et al., 2023) and auto-regressive prediction (Radford et al., 2018) for feature learning.

Moreover, we also consider the distinctions between events and language, primarily in information density and spatial locality. One token in NLP has significant semantic meaning, whereas an individual event token offers limited information, gaining significance only when aggregated over time. This steers our focus towards the aggregated global information rather than isolated event instances. Additionally, events typically signify spatially localized changes, contrasting with language, which lacks inherent spatial attributes. This motivates modeling correlations primarily between spatially adjacent events to reduce complexity.

In this work, we introduce a new A2S framework with a more expressive asynchronous encoder, which achieves event-by-event feature updating, and a self-supervised method for generalizable feature learning (Figure 1). The asynchronous encoder is built on the basis of LA, which enables recurrent inference and parallel training. We propose to use the matrix-value hidden state (MVHS) introduced in LA as output to expand the model's expressivity on the aggregated global information and propose patch-wise encoding to leverage the locality. For ease of implementation and better performance, our model is built on an open source high-performance LA architecture RWKV-6 (Peng et al., 2024), which implements MVHS and is stable to train.

We propose a self-supervised learning (SSL) method to learn generalizable features, which consists of two tasks: multi-representation prediction (MRP) and next-representation prediction (NRP). The former encourages the feature to learn diverse information from multiple handcrafted (converted) representations (Lagorce et al., 2016; Sironi et al., 2018) comprehensively, and the latter, motivated by next-token prediction (NTP) in NLP (Radford et al., 2018), pushes the model beyond simple memorization. Our framework is superior to the previous A2S work (Martin-Turrero et al., 2024) on recognition tasks, and it is the first A2S framework, to the best of our knowledge, that successfully masters demanding detection tasks, achieving a 0.477 mAP on the Gen1 dataset. We name our framework as **EV**ent **A**synchronous feature learning (EVA). Our contributions include:

1. An asynchronous LA-based encoder architecture derived from RWKV-6, enabling efficient event-by-event feature updating with improved expressivity.

2. A novel multi-task SSL method learning features generalizable to various downstream tasks.

3. We present EVA, a new A2S framework that combines the asynchronous encoder with self-supervised feature learning, outperforming the previous A2S work on recognition tasks and, for the first time, successfully masters demanding detection tasks with a 0.477 mAP on the Gen1 dataset.

## 2 RELATED WORK

### 2.1 ASYNCHRONOUS EVENT PROCESSING

To leverage the sparsity and high temporal resolution of event data, several recent studies have focused on asynchronous event processing. These include convolutional methods (Messikommer

et al., 2020; Cannici et al., 2019), graph-based models (Li et al., 2021; Schaefer et al., 2022; Dalgaty et al., 2023; Dampfhoffer et al., 2025; Gehrig & Scaramuzza, 2024; Li et al., 2025), attention-based mechanisms (Kamal et al., 2023), LSTM-based solutions (Cannici et al., 2020; Santambrogio et al., 2024), and point cloud approaches (Sekikawa et al., 2019; Martin-Turrero et al., 2024). Convolutional methods derive local activation update rules in standard (Cannici et al., 2019) or sparse (Messikommer et al., 2020) convolution layers. While they can leverage sparsity and reduce computation, they discard the temporal information. Graph-based methods transform events into spatial-temporal graphs to capture both spatial and temporal information and leverage sparsity. However, graph-based methods show limitations in temporal accumulation (Dampfhoffer et al., 2025). An attention-based, memory-augmented method was proposed in Kamal et al. (2023) to process batched events at each timestep. The LSTM-based Matrix-LSTM (Cannici et al., 2020) method learns and recurrently updates a representation with a pixel-wise LSTM for downstream tasks. FARSE-CNN (Santambrogio et al., 2024) integrates LSTMs within the neurons of convolutional layers to capture spatio-temporal information and employs local update rules. Our work is most related to the A2S method point cloud-based ALERT-Transformer (Martin-Turrero et al., 2024), which uses EventNet (Sekikawa et al., 2019) for asynchronous feature encoding. However, EventNet does not perform hierarchical learning and its expressivity is therefore limited. Our method follows the same A2S paradigm but improves expressivity with a new architecture.

## 2.2 LANGUAGE-LIKE EVENT PROCESSING

Recently, several works studied processing raw events in a token-wise way. Jiang et al. (2022) processes raw event sequence with Transformer directly, but is limited to the quadratic complexity of softmax attention on long event sequences. To solve this, Schöne et al. (2024) and Soydan et al. (2024) introduce state space models (SSMs) with linear complexity for asynchronous event processing. However, these methods do not discuss the similarity between events and language. Concurrent work (Fang & Panda, 2025) also models raw events with linear attention (Peng et al., 2023a). However, our work focuses on the improvement of feature expressivity under the A2S paradigm, while Fang & Panda (2025) focuses on learning event representations like Word2Vec (Church, 2017).

## 2.3 SELF-SUPERVISED LEARNING ON EVENT DATA

Event data SSL can be broadly categorized into methods that learn from events only and those that incorporate other modalities. For event-only SSL, MEM (Klenk et al., 2024) uses the masked image modeling method (Bao et al., 2021) to event images for pretraining. Other works (Barchid et al., 2023; Yang et al., 2024) utilize contrastive learning with joint embedding architectures (Chen et al., 2020; He et al., 2020) on event images. These methods typically convert events to images and adapt established image SSL methods to the event domain. For SSL with other modalities, events are often paired with intensity images. In Yang et al. (2023b), contrastive pretraining is explored on event data synthetically converted from RGB images and their source images. EVRepSL learns representations by exploiting the relationship between paired images and events (Qu et al., 2024).

## 3 METHOD

Our framework leverages the event-language analogy, and we first analyze their similarities and distinctions. On one hand, events and language are similar in their (i) *sequential nature* and (ii) *incremental manner*. Events convey local intensity changes, similar to language tokens that incrementally build the context (Figure 2). On the other hand, key distinctions exist in their (i) *information density* and (ii) *spatial locality*. An individual language token has explicit semantics. In contrast, events only record pixel-level intensity changes and require temporal aggregation to become informative.

As illustrated in Figure 3, our A2S framework first transforms raw events into tokens and embeds them with their spatial and temporal attributes. These embeddings are then asynchronously encoded into features by the RWKV-6-based encoder. The features are learned via SSL during training and are generated event-by-event at inference for real-time downstream applications.

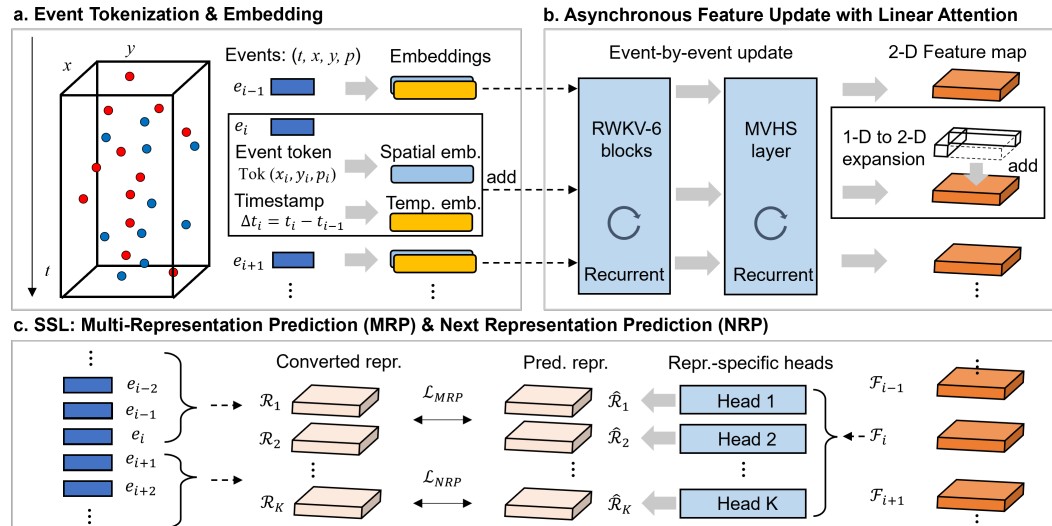

Figure 2: Parallels between events and language: their sequential nature and incremental manner.

Figure 3: Architecture of the A2S framework: (a) Tokenizing and embedding events. (b) Asynchronous encoding via LA. (c) MRP and NRP for self-supervised feature learning.

## 3.1 ARCHITECTURE: ASYNCHRONOUS ENCODER

Key challenges in designing an asynchronous event encoder include: (i) allowing recurrent inference for real-time feature updates and parallel training on long event sequences, (ii) continuously encoding events with variable temporal intervals, and (iii) efficiently and expressively aggregating information into the feature. To address the first challenge, drawing from the event-language analogy, we find that recent linear attention (LA) supports both recurrent inference and parallel training. Building on LA, we incorporate temporal embedding, MVHS as output, and patch-wise encoding for the other challenges.

### 3.1.1 EVENT TOKENIZATION AND EMBEDDING

We define an event sequence as $\mathcal{E} = \{e_i = (t_i, x_i, y_i, p_i), i = 1, 2, \dots\}$, where $t, x, y, p \in \mathbb{R}$ are the timestamp, coordinates, and polarity of events, with $p \in \{0, 1\}$ indicating the two polarities. The spatial component of each event is tokenized using a bijection Tok as:

$$\begin{aligned}
\text{Tok} : \mathcal{E} &\to \mathbb{N} \\
(x, y, p) &\mapsto p \times H \times W + y \times W + x,
\end{aligned} \tag{1}$$

where $H, W$ denote the height and width of the spatial area containing the event sequence. The vocabulary size is thus $2 \times H \times W$. This mapped token represents the spatial location of the event.

Each tokenized event $e_i = (t_i, \text{Tok}(x_i, y_i, p_i))$ is then embedded as an input vector $\boldsymbol{x}_i \in \mathbb{R}^D$ to the encoder. For event tokens that represent spatial locations, we employ a learnable embedding layer on the tokens and get $\text{Emb}_{\text{spatial}}(e_i) \in \mathbb{R}^D$.

Events are inherently asynchronous, so temporal information must be embedded in the input. Considering that event cameras generate events at high temporal resolution, the absolute timestamp (in

$\mu s$) quickly grows large in continuous operation; embedding the absolute timestamp would push the model beyond the range observed during the pretraining period, leading to the same kind of length-extrapolation failures in language models (Press et al., 2021; Su et al., 2024; Chen et al., 2023). We therefore embed the time difference $\Delta t_i = t_i - t_{i-1}$ instead of the absolute timestamp, and use sinusoidal embedding (Vaswani et al., 2017). The final embedding is then formed by summing the spatial and temporal embeddings.

### 3.1.2 MODELING EVENT SEQUENCE WITH LINEAR ATTENTION

**Background: Linear attention in RWKV-6.** The asynchronous encoder is built on RWKV-6, a high-performance linear attention architecture for language modeling(Peng et al., 2024). For an input sequence $\{\boldsymbol{x}_i \in \mathbb{R}^{D \times 1}\}_{i=1}^T$ of length $T$, a single RWKV-6 block performs token mixing (TM) and channel mixing (CM).

In TM, inputs are first mapped to $\{\boldsymbol{r}_i, \boldsymbol{w}_i, \boldsymbol{k}_i, \boldsymbol{v}_i \in \mathbb{R}^{D \times 1}\}_{i=1}^T$ and subsequently mixed by the $\boldsymbol{rwkv}$ operator:

$$\boldsymbol{y}_i = \left[ \sum_{t=1}^{i-1} \mathrm{diag} \left( \bigotimes_{j=t+1}^{i-1} \boldsymbol{w}_j \right) \boldsymbol{k}_t \boldsymbol{v}_t^{\mathrm{T}} + \mathrm{diag}\left(\boldsymbol{u}\right) \boldsymbol{k}_i \boldsymbol{v}_i^{\mathrm{T}} \right] \boldsymbol{r}_i \in \mathbb{R}^{D \times 1}, \tag{2}$$

where $\boldsymbol{u} \in \mathbb{R}^{D \times 1}$ is a learnable parameter. This $\boldsymbol{rwkv}$ operator involves only element-wise prefix summations and products and therefore enables parallel training with scan (Martin & Cundy, 2018). Equation 2 can be expressed in a recurrent form:

$$\begin{aligned} \boldsymbol{S}_i &= \mathrm{diag}\left(\boldsymbol{w}_i\right) \boldsymbol{S}_{i-1} + \boldsymbol{k}_i \boldsymbol{v}_i^{\mathrm{T}} \in \mathbb{R}^{D \times D}, \\ \boldsymbol{y}_i &= \left( \boldsymbol{S}_{i-1} + \mathrm{diag}\left(\boldsymbol{u}\right) \boldsymbol{k}_i \boldsymbol{v}_i^{\mathrm{T}} \right) \boldsymbol{r}_i \in \mathbb{R}^{D \times 1}. \end{aligned} \tag{3}$$

Consequently, the hidden state $\boldsymbol{S}$ can be updated recurrently at inference upon the arrival of each new event, allowing efficient training and recurrent inference on continuous long event sequences.

To better capture diverse information aspects, RWKV-6 uses multi-head TM. Each input is split into $N$ heads of dimension $D_{\mathrm{head}} = D/N$, and the $\boldsymbol{rwkv}$ operator acts on each head individually. The hidden state in multi-head TM is $\boldsymbol{S} \in \mathbb{R}^{N \times D_{\mathrm{head}} \times D_{\mathrm{head}}}$.

The CM layer transforms TM outputs using a feed-forward network, similar to Transformers (Vaswani et al., 2017). Appendix B provides details.

**Matrix-value hidden states (MVHS) as output.** NLP tasks typically involve sequence-to-sequence models mapping 1-D embeddings to 1-D embeddings. We propose to train directly on the 2-D MVHS $\boldsymbol{S}$, rather than the 1-D output $\boldsymbol{y}$, and use $\boldsymbol{S}$ as the event feature, which offers several benefits. First, MVHS, as the hidden state, naturally contains aggregated information. This aggregated feature is precisely what we want, instead of the mapping from embeddings to embeddings in standard auto-regressive pretraining (our attempt to predict the next single event embedding failed to converge). Secondly, MVHS provides an expanded memory of size $N \times D_{\mathrm{head}} \times D_{\mathrm{head}}$ without increasing model width $D$, thus improving the expressivity of the learned feature and enabling a lightweight architecture for real-time event processing.

Using MVHS as output can reduce model size by approximately $D_{\mathrm{model}}/N$ versus 1-D outputs (Appendix G). Moreover, the 2-D structure of MVHS helps to learn fine-grained spatial features, improving performance over prior 1-D features (Sekikawa et al., 2019; Martin-Turrero et al., 2024).

### 3.1.3 PATCH-WISE ENCODING OF FEATURES

To exploit the locality of the event, we follow the practice in Martin & Cundy (2018) and use Patch-wise encoding (PWE) for the feature. Specifically, for an event camera with resolution $(H_{\mathrm{sensor}}, W_{\mathrm{sensor}})$ and patch size $P$, we separate events to $H_{\mathrm{sensor}} \times W_{\mathrm{sensor}}/P^2$ sequences by their coordinates. As Figure 4 illustrates, we encode a feature for each patch separately. With PWE, the model

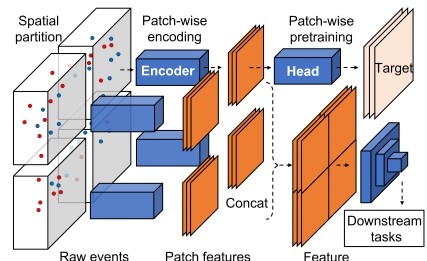

Figure 4: Patch-wise feature encoding. Events are partitioned and encoded by patch. The patch feature could be concatenated for downstream tasks.

size of the encoder can be reduced by a factor of approximately the number of patches ($\# \, patches$), which helps real-time inference (see Appendix G for discussion). Furthermore, since the asynchronous encoder trains on fixed-size patches, this patch-wise encoding allows the application to event cameras with different resolutions.

## 3.2 Self-supervised feature learning on asynchronous events

We employ SSL to train the asynchronous encoder. Since this training procedure is task-irrelevant, the resulting event feature is not tailored to a certain task, thereby exhibiting better generalizability. The proposed SSL method consists of two tasks: multi-representation prediction (MRP) and next representation prediction (NRP).

### 3.2.1 Multi-representation prediction

For MRP, we force the encoded feature to predict multiple handcrafted (converted) event representations (Lagorce et al., 2016; Sironi et al., 2018; Innocenti et al., 2021), such as event count (EC) and time surface (TS), to learn a more comprehensive feature. For events $\mathcal{E} = \{e_i = (t_i, x_i, y_i, p_i), i = 1, 2, \dots\}$, we expect the asynchronous encoder $\mathcal{M}_{\boldsymbol{\theta}}$ parametrized by $\boldsymbol{\theta}$ to learn an event-by-event feature as $\mathcal{F}_i = \mathcal{M}_{\boldsymbol{\theta}}(\{e_j\}_{j \leq i})$. Given $K$ handcrafted representations $\{\mathcal{R}^k\}_{k=1}^{K}$, we predict these representations from $\mathcal{F}$ with $K$ specific heads parameterized by $\boldsymbol{\Theta} = \{\boldsymbol{\theta}_k\}_{k=1}^{K}$. The objective is

$$\arg\max_{\boldsymbol{\theta}, \boldsymbol{\Theta}} \mathbb{E}_i \prod_{k=1}^{K} \mathbf{Pr}\left(\mathcal{R}_i^k | \mathcal{F}_i = \mathcal{M}_{\boldsymbol{\theta}}\left(\{e_j\}_{j \leq i}\right); \boldsymbol{\theta}_k\right), \qquad (4)$$

where $\mathbf{Pr}$ means probability. MRP is motivated by the insight that different handcrafted representations capture different aspects of information about the raw events. They provide the required information for various tasks. Therefore, we anticipate that learning from multiple converted representations will yield a feature generalizable to diverse downstream tasks.

### 3.2.2 Next representation prediction

Inspired by auto-regressive pretraining of language models, where models learn by next-token prediction (Radford et al., 2018), NRP forces the encoded feature to predict handcrafted representations for a future time interval. For a future time window $\Delta T$, the objective of NRP is

$$\arg\max_{\boldsymbol{\theta}, \boldsymbol{\Theta}'} \mathbb{E}_i \prod_{k=1}^{K'} \mathbf{Pr}\left(\mathcal{R}^k\left(\{e | t_i < t(e) \leq t_i + \Delta T\}\right) | \mathcal{F}_i = \mathcal{M}_{\boldsymbol{\theta}}\left(\{e_j\}_{j \leq i}\right); \boldsymbol{\theta}_k'\right). \qquad (5)$$

We aim for the model to understand inherent motion patterns and go beyond just memorizing the history by learning to predict the future. In particular, individual events are ineffective as prediction targets due to their limited information and unpredictable noise. In contrast, converted representations are more informative and noise-robust, offering a more reliable supervision signal.

## 4 Experiments

### 4.1 Object recognition

**Datasets.** To compare with previous A2S work (Martin-Turrero et al., 2024), we evaluate our EVA on two event-based recognition tasks: action recognition (DVS128-Gesture (Amir et al., 2017)) and binary classification (N-Cars (Sironi et al., 2018)). DVS128-Gesture provides human actions recorded by the DVS128 event camera, containing 1342 files of around 6 seconds from 29 subjects under 3 different lighting conditions. N-Cars provides a benchmark for the binary classification task of car recognition, containing 24029 files of 100 ms recorded by the ATIS event camera.

**Implementations.** We present an asynchronous encoder with embedding width $D = 128$, 3 RWKV-6 blocks of width $D = 128$, and 1 MVHS layer as output to generate the learned feature. For PWE, we use $P = 16$. On DVS128-Gesture, to reduce the latency of downstream classification, we use $D_{\text{head}} = 8$ for a $2\times$ shrink in height and width of the feature and use a lightweight 14-layer ResNet (He et al., 2016) (referred to as ResNet-14) for classification. On N-Cars, however, since the samples

are cropped to small sizes (Sironi et al., 2018), we also present an encoder with $D_{head} = 16$ for a full height and width feature. To maintain the feature size, we use only half of the theoretical 8 channels of the MVHS output. The model implementation details are given in Appendix D.

On DVS128-Gesture, the encoder is pretrained on event sequences of length 8192 with batch size 32 and Adam optimizer (Kingma & Ba, 2014) with learning rate 0.001. It takes 10 hours to train 100 epochs with an RTX 3090 GPU. On N-Cars, the encoder is pretrained with a sequence length of 512 with batch size 32 and learning rate 0.001. It takes 1 hour to train 20 epochs on an RTX 3090 GPU. We use EC and TS for MRP and EC for NRP as the target. To balance the multi-task losses, we use uncertainty weights (Kendall et al., 2018). The details of loss items are given in Appendix E.

**Metrics.** On DVS128-Gesture, following previous work (Martin-Turrero et al., 2024), we evaluate the model on samples of length 8192 and report the accuracy on samples (sample accuracy, SA) and the accuracy after majority voting on all the samples of each file (file voting accuracy, FVA). We also report the complexity (MAC) and inference latency on the RTX 3090 GPU. On N-Cars, we report accuracy.

**Results.** On DVS128-Gesture, our model achieves 96.9% FVA and 92.9% SA (see Table 1), outperforming the best result in previous A2S work (94.1% FVA and 84.6% SA) by 2.8% and 8.3% respectively. Compared with ALERT-Tr. (+RM), our EVA achieves better performance with a smaller parameter size and complexity. ALERT-Tr. (+LMM) indeed has fewer parameters, but with a much lower accuracy of 72.6% in SA, 20% less accuracy than ours. It is worth noting that our EVA, with a lightweight ResNet classifier, has much lower inference time (1.5 ms) compared with both the LMM and RM. Our asynchronous encoder has a larger inference latency compared with previous work because of its hierarchy learning architecture. However, it has a shorter processing time for an 8192-length sample, which is about 100 ms in DVS128-Gesture. On N-Cars (Table 2), previous A2S work (Martin-Turrero et al., 2024) did not report the results with their best model (RM), only gave the accuracy of LMM to show its low complexity. Therefore, here we additionally compare our EVA with (i) dense event image-based methods (Peng et al., 2023b; Klenk et al., 2024) and (ii) methods that learn the representation/feature from raw events (Gehrig et al., 2019; Cannici et al., 2020). In comparison with ALERT-Tr. (+LMM), our method has better accuracy and less classifier latency. Since the files in N-Cars are small in number of events and the pretraining on this dataset is not sufficient, we also present the result with features generated by an encoder pretrained on Gen1 for detection tasks (EVA-L in Table 3). For a fair comparison with Gehrig et al. (2019) and Cannici et al. (2020), we follow their practice and use ResNet-34 pretrained on ImageNet (Deng et al., 2009) as the classifier. Our model, with the encoder pretrained on Gen1 and ResNet-34, has an accuracy of 96.3%, higher than the methods with learned representations.

Table 1: Results on DVS128-Gesture. Latency refers to the time to process a sample of length 8192. We compare with previous A2S work ALERT-Tr. (Martin-Turrero et al., 2024).

| Model | # Params | MAC per | | Latency | Acc. | |
| | | Event | Sample | | SA | FVA |
|---|---|---|---|---|---|---|
| ALERT-Tr. | 1.41 M | 1.22 M | 8.83 G | 5.8 ms | 84.6% | 94.1% |
| (+RM) | 13.96 M | 1.30 M | 9.42 G | 9.6 ms | | |
| ALERT-Tr. | 0.04 M | 0.0040 M | 0.03 G | **3.9** ms | 72.6% | 89.2% |
| (+LMM) | 0.57 M | 0.0074 M | 0.06 G | 6.0 ms | | |
| EVA (ours) | 0.62 M | 0.60 M | 4.79 G | 14.7 ms | **92.9%** | **96.9%** |
| (+ResNet-14) | 2.83 M | 0.28 M | 1.82 G | **1.5** ms | | |

## 4.2 Object detection

**Dataset.** We evaluate our work on the challenging automotive detection dataset Gen1 (De Tournemire et al., 2020), which includes 39 hours of data recorded by the Gen1 event camera with a sensor resolution of $304 \times 240$.

Table 2: Results on N-Cars. EVA is pretrained on N-Cars, and EVA-L is pretrained on Gen1.

| Model | Event Repr. | Acc. |
|---|---|---|
| GET (Peng et al., 2023b) | Event count $\rightarrow$ Token | 96.9% |
| MEM (Klenk et al., 2024) | Event count | **98.4%** |
| EST (Gehrig et al., 2019) | Learned | 92.5% |
| Matrix-LSTM (Cannici et al., 2020) | Learned | 95.8% |
| ALERT-Tr. (+LMM) | Learned | 85.6% |
| EVA (+ResNet-14) (ours) | Learned | 91.4% |
| EVA (+ResNet-14) ($D_{head} = 16$) (ours) | Learned | 91.6% |
| EVA-L (+ResNet-34) ($D_{head} = 16$) (ours) | Learned | 96.3% |

**Implementations.** We present EVA of 3 RWKV-6 blocks, one MVHS layer with $D = 128$. We also present a larger model with $D = 192$ (referred to as EVA-L). We use patch size $P = 16$ and $D_{head} = 16$ to keep the spatial resolution. Like N-Cars, we use only half of the output channels of the MVHS layer. For the downstream detection task, we use RVT-B (Gehrig & Scaramuzza, 2023), a lightweight state-of-the-art (SOTA) Transformer-based synchronous backbone and YOLOX detection head (Ge et al., 2021) on the learned features. The model implementation details are given in Appendix D.

We pretrain the encoder with sequence length $T = 2048$ and batch size 64 on 4 RTX 3090 GPUs. The learning rate is initially 0.001 with a decay of $\gamma = 0.9$ every epoch. We use EC and TS for MRP objectives and EC for NRP. The details are given in Appendix E.

**Results.** Existing A2S work did not report results on object detection tasks. Therefore, here we compare our method with (i) synchronous dense methods and (ii) end-to-end asynchronous methods. The results are shown in Table 3. Asynchronous methods have minimal latency but lower accuracy compared with synchronous methods. Our best model achieves a 47.7 mAP, and improves the SOTA synchronous method (Gehrig & Scaramuzza, 2023) with a lower number of channels of the input feature (6 for our EVA compared with 20 for RVT-B). This is the first result by A2S methods on event-based detection tasks, demonstrating the effectiveness of our method.

Table 3: Object detection performance on Gen1. A refers to end-to-end asynchronous methods. S refers to synchronous dense methods.

| Model | Type | mAP (%) |
|---|---|---|
| NVS-S (Li et al., 2021) | A | 8.6 |
| AEGNN (Schaefer et al., 2022) | A | 14.5 |
| Asynet (Messikommer et al., 2020) | A | 16.3 |
| FARSE-CNN (Santambrogio et al., 2024) | A | 30.0 |
| DAGr-L (Gehrig & Scaramuzza, 2024) | A | 32.1 |
| ASTMNet (Li et al., 2022) | S | 46.7 |
| HMNet-L3 (Hamaguchi et al., 2023) | S | 47.1 |
| RVT-B (Gehrig & Scaramuzza, 2023) | S | 47.2 |
| GET (Peng et al., 2023b) | S | **47.9** |
| EVA (+RVT-B) (ours) ($D = 128, D_{head} = 16$) | A2S | 47.5 |
| EVA-L (+RVT-B) (ours) ($D = 192, D_{head} = 16$) | A2S | 47.7 |

## 4.3 TIMING EXPERIMENTS

**MACs. per event.** We timed our EVA models, implemented in Python and CUDA, on an RTX 3090 GPU. We summarize the results in Table 4 and the event rate (number of events per second) of the datasets in Table 5. For DVS128-Gesture and N-Cars, the event rates are both within 100,000 per second, lower than the throughput of the EVA models (at least 541,000 per second). For Gen1 with a larger resolution, the event rate surpasses the throughput of our EVA-L model. Since patch-

wise encoding (PWE) is used in our EVA encoder, the features of each patch could be encoded independently. Therefore, considering that the patch event rate of the Gen1 dataset is 2.17 K/s, much lower than the throughput of EVA-L, the real-time encoding on Gen1 could still be achieved.

Table 4: Throughput and MACs of EVA models.

| Model | # Param | MACs per Event | Throughput (M/s) |
|---|---|---|---|
| EVA | 0.6 M | 0.60 M | 0.611 |
| EVA($D_{head} = 16$) | 0.6 M | 0.61 M | 0.560 |
| EVA-L | 1.4 M | 1.31 M | 0.541 |

Table 5: Event rates of different datasets.

| Dataset | Camera | Resolution | Event rate (M/s) | Patch event rate (K/s) |
|---|---|---|---|---|
| DVS128-Gesture | DVS128 | $128\times 128$ | 0.058 | 0.90 |
| N-Cars | ATIS | $100\times 120$ | 0.039 | 4.07 |
| Gen1 | ATIS | $304\times 240$ | 0.618 | 2.17 |

**Latency vs. Context length.** We evaluate the inference time with respect to input context length. As shown in Figure 5, the inference time is basically proportional to the context length (number of input events). We could leverage temporal subsampling (Santambrogio et al., 2024) or pooling (Soydan et al., 2024) to reduce MACs, which are commonly used in asynchronous event processing.

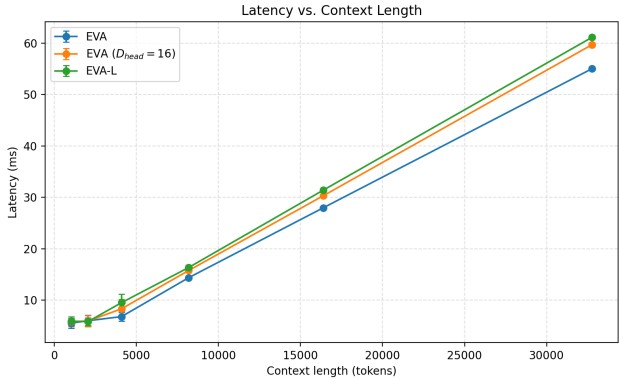

Figure 5: Inference latency on different sequence lengths.

**Latency vs. Sensor resolution.** For high-resolution event cameras (i.e., Gen 3 with $1280 \times 720$ pixels), the event rate would increase proportionally to the sensor size. Therefore, these high-frequency event sequences would challenge our EVA encoder. Theoretically, since EVA conducts a PWE strategy, the features of each patch could be computed in parallel. The overall runtime is only determined by the longest computation time among all the patches, and thus only determined by the patch event rate. From Table 5, the patch event rate is stable across event cameras with different resolutions. Therefore, our method could, in principle, be applicable to event cameras at any resolution, without introducing further latency.

## 4.4 ABLATION STUDY

We ablate architecture components (MVHS and temporal embedding) and SSL objectives to check their contribution. The effect of patch size is also investigated. The ablation is conducted on DVS128-Gesture, following the implementations in the object recognition experiment. We report the loss items of SSL during pretraining and downstream accuracy on the validation set.

**Ablation of MVHS and temporal embedding.** To remove MVHS, we keep the model width $D = 128$ unchanged and let $D_{\text{head}} = 1$ in the output layer. The output becomes a vector with shape

$128 \times 1 \times 1$. The results are shown in Tab 6. Models without MVHS or temporal embedding show a decrease in both FVA and SA, and also have a larger pretraining loss.

Table 6: Ablation of MVHS and temporal embedding.

| MVHS | Tempo. Emb. | MRP Loss | | NRP Loss | Acc. | |
|---|---|---|---|---|---|---|
| | | EC ($\downarrow$) | TS ($\downarrow$) | EC($\downarrow$) | FVA ($\uparrow$) | SA ($\uparrow$) |
| ✓ | ✓ | **0.366** | **$3.16 \times 10^{-3}$** | **0.767** | **98.1%** | **94.7%** |
| ✓ | | 1.870 | $4.56 \times 10^{-2}$ | 0.933 | 87.8% | 81.1% |
| | ✓ | 0.826 | $9.34 \times 10^{-3}$ | 0.834 | 97.4% | 94.1% |

**Ablation of SSL objective.** We remove the SSL objectives to show their contribution to the learned feature. Results in Table 7 show that removing the items of SSL decreases the performance of the model. One interesting thing is that learning only one representation, such as EC, will not contribute to a smaller pretraining loss compared with learning all the representations. This indicates that learning one single representation could benefit from learning other representations.

Table 7: Ablation of objectives of MRP and NRP.

| MRP | | NRP | MRP Loss | | NRP Loss | Acc. | |
|---|---|---|---|---|---|---|---|
| EC | TS | EC | EC ($\downarrow$) | TS ($\downarrow$) | EC($\downarrow$) | FVA ($\uparrow$) | SA ($\uparrow$) |
| ✓ | ✓ | ✓ | **0.366** | **$3.16 \times 10^{-3}$** | **0.767** | **98.1%** | **94.7%** |
| ✓ | ✓ | | 0.639 | $1.03 \times 10^{-2}$ | – | 96.8% | 94.4% |
| ✓ | | | 0.701 | – | – | 96.8% | 91.7% |
| | ✓ | | – | $1.40 \times 10^{-2}$ | – | 97.4% | 91.6% |
| | | ✓ | – | – | 0.825 | 96.8% | 91.8% |

**Effect of patch size.** We change the patch size from 16 to 128 (w/o patching in this case), while keeping the width of the encoder $D = 128$ and the sequence length $T = 8192$ for different patch sizes. The results in Table 8 show that a small patch size gives better accuracy. Patch size $P = 128$ gives a smaller pretraining loss because the spatial sparsity makes a larger patch have more blank areas.

Table 8: Effect of the choice of patch size

| Patch Size | MRP Loss | | NRP Loss | Acc. | |
|---|---|---|---|---|---|
| | EC ($\downarrow$) | TS ($\downarrow$) | EC($\downarrow$) | FVA ($\uparrow$) | SA ($\uparrow$) |
| 16 | 0.366 | **$3.16 \times 10^{-3}$** | 0.767 | **98.1%** | **94.7%** |
| 32 | 0.454 | $1.67 \times 10^{-2}$ | 0.393 | 94.8% | 91.0% |
| 64 | 0.233 | $1.45 \times 10^{-2}$ | 0.149 | 95.5% | 89.6% |
| 128 (w/o patching) | **0.108** | $9.88 \times 10^{-3}$ | **0.073** | 97.4% | 89.3% |

## 5 CONCLUSION

We introduce a new A2S framework, EVA, for real-time event camera data processing. The framework consists of an efficient and expressive linear attention-based asynchronous encoder for event-by-event feature updating and an SSL method to learn a generalizable event feature. EVA outperforms the previous A2S work (Martin-Turrero et al., 2024), achieves remarkable performance on event-based detection tasks, and learns generalizable features applicable in diverse downstream tasks, demonstrating its potential towards real-time event-based applications.

The limitations of our work are discussed in Appendix H.

## REPRODUCIBILITY STATEMENT

**Code**: Code is available `https://github.com/haohq19/eva`. **Datasets**: The data pre-processing steps are described in Appendix C. **Experiments**: All experimental settings, including hyperparameters and model architectures, are detailed in Section 4 and Appendix D. The random seeds used for all experiments are fixed.

## ACKNOWLEDGEMENT

This work was supported by STI 2030-Major Projects 2021ZD0200300 and High Performance Computing Center of Tsinghua University.

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

## A    EXTENDED EXPERIMENTAL RESULTS

### A.1    OBJECT RECOGNITION

We provide object recognition experiments on static-captured event camera datasets. We evaluate the EVA feature on N-Caltech101 (Orchard et al., 2015) dataset, which is created by moving an event camera that is focused on a screen displaying the original Caltech101 dataset samples. The dataset consists of 100 object classes and an additional background class. Since the dataset does not provide an official train/test split, we follow the split in EST (Gehrig et al., 2019) for fair comparison.

The results are shown in Table 9. Our EVA method outperforms all the asynchronous methods and has competitive accuracy compared with synchronous methods. Our method surpasses EST (Gehrig et al., 2019) and MatrixLSTM (Cannici et al., 2020), which learn representations in a supervised, end-to-end manner. E2VID reconstructs gray-scale images and introduces an extra latency (Rebecq et al., 2019). Our EVA method, however, does not lead to further latency and only shows a marginal 0.3% drop in accuracy. This demonstrates that our EVA feature could also be used on the challenging static-captured datasets.

Table 9: Object recognition results on N-Caltech101. A. refers to asynchronous. S. refers to synchronous.

| Model | Type | Acc. |
|---|---|---|
| HOTS (Lagorce et al., 2016) | A. | 21.0 % |
| NVS-S (Li et al., 2021) | A. | 67.0 % |
| AEGNN (Schaefer et al., 2022) | A. | 66.8 % |
| Asynet (Cannici et al., 2019) | A. | 74.5 % |
| FARSE-CNN (Santambrogio et al., 2024) | A. | 68.7 % |
| EST (Gehrig et al., 2019) | S. | 83.7 % |
| Matrix-LSTM (Cannici et al., 2020) | S. | 84.3 % |
| Pillar (Fan et al., 2025) | S. | 85.3 % |
| E2VID (Rebecq et al., 2019) | S. | **86.6** % |
| EVA-L (ours) | A2S | 86.3 % |

## A.2 OBJECT DETECTION

To better demonstrate the expressivity and low-latency advantage of the learned asynchronous EVA feature, here we compare our EVA method with other event-representing methods under an identical neural network backbone (Swin Transformer V2). We compare both accuracy and representing latency, which refers to the time required for computing the representation when a new event arrives (Fan et al., 2025). The results are shown in Table 10.

Table 10: Comparison of different event representations with an identical Swin V2 backbone on the Gen1 detection task. Representing latency refers to the time required for computing the representation when a new event arrives.

| Method | mAP (%) | Asynchronous | Representing latency (ms) |
|---|---|---|---|
| Voxel(Fan et al., 2025) | 39.5 | | 16.3 |
| ERGO-12(Zubić et al., 2023) | 50.4 | | > 10 |
| Pillar(Fan et al., 2025) | **53.1** | | 11.4 |
| HOTS(Lagorce et al., 2016) | 49.0 | ✓ | 1.0 |
| TORE(Baldwin et al., 2022) | 43.6 | ✓ | < 0.1 |
| EVA-L (ours) | 49.7 | ✓ | < 0.1 |
| EVA (ours) | 49.0 | ✓ | < 0.1 |

Synchronous representations provide better performance, but with additional latency. Our EVA method outperforms all the asynchronous representations and maintains its low-latency characteristic. This further demonstrates the advantages of our method in low-latency applications.

## A.3 TIMING EXPERIMENTS

We also evaluate the accuracy with different event data downsampling rates. Considering that in real-world applications, an event camera could generate events at different rates depending on the scene dynamics, and in case of high event rates which surpass the throughput of the EVA encoder, it is critical to understand how it performs under such conditions. We conduct experiments on DVS128-Gesture dataset, and downsample the raw event sequence to lower event rates. We evaluate

the recognition accuracy and throughput of our EVA method. We freeze the pretrained weights of EVA encoder. The results are shown in Table 11. For a shorter sequence length, the throughput correspondingly increases, while the accuracy will decrease due to the information loss from downsampling. However, even with a $8\times$ downsampling, our accuracy (95.1%) is still higher than the baseline (ALERT-Tr., 94.1%), with a higher throughput (4888 K/s vs 1412 K/s).

Table 11: Accuracy and throughput under different event rates with downsampling.

| Model | Length | SA. | FVA. | Throughput (K/s) |
|---|---|---|---|---|
| | 8192 | 92.9 | 96.9 | 611 |
| | 4096 | 91.8 | 96.2 | 1222 |
| EVA (ours) | 2048 | 91.3 | 95.3 | 2444 |
| | 1024 | 90.0 | 95.1 | 4888 |
| | 512 | 88.4 | 93.4 | 9776 |
| ALERT-Tr. (+RM) | 8192 | 84.6 | 94.1 | 1412 |
| ALERT-Tr. (+LMM) | 8192 | 72.6 | 86.2 | 2100 |

## B   ARCHITECTURE DETAILS OF THE ASYNCHRONOUS ENCODER

**RWKV-6 details.** As described in (Peng et al., 2024), one RWKV-6 block is composed of token mixing (TM) and channel mixing (CM). For input sequence $\{\boldsymbol{x}_i \in \mathbb{R}^D\}_{i=1}^T$ of length $T$, TM first conduct a data-dependent linear interpolation (ddlerp) between $\boldsymbol{x}_i$ and $\boldsymbol{x}_{i-1}$ by

$$
\begin{aligned}
&\mathrm{lora}_{\{r,k,v,g\}}(\boldsymbol{x}_i) = \boldsymbol{\lambda}_{\{r,k,v,g\}} + \tanh(\boldsymbol{x}_i \boldsymbol{A}_{\{r,k,v,g\}})\boldsymbol{B}_{\{r,k,v,g\}}, \\
&\mathrm{ddlerp}_{\{r,k,v,g\}}(\boldsymbol{x}_i, \boldsymbol{x}_{i-1}) = \boldsymbol{x}_i + (\boldsymbol{x}_{i-1} - \boldsymbol{x}_i) \odot \mathrm{lora}_{\{r,k,v,g\}}(\boldsymbol{x}_i + (\boldsymbol{x}_{i-1} - \boldsymbol{x}_i) \odot \boldsymbol{\mu}),
\end{aligned}
\tag{6}
$$

where $\boldsymbol{\mu}, \boldsymbol{\lambda}_{\{r,k,v,g\}} \in \mathbb{R}^D$ are learnable parameters and $\boldsymbol{A}_{\{r,k,v,g\}} \in \mathbb{R}^{D \times D_{\mathrm{lora}}}, \boldsymbol{B}_{\{r,k,v,g\}} \in \mathbb{R}^{D_{\mathrm{lora}} \times D}$ are trainable weight matrices. The vectors in Equation 2 are calculated as

$$
\begin{aligned}
\boldsymbol{r}_t &= \mathrm{ddlerp}_r(\boldsymbol{x}_i, \boldsymbol{x}_{i-1})\boldsymbol{W}_r, \\
\boldsymbol{k}_t &= \mathrm{ddlerp}_k(\boldsymbol{x}_i, \boldsymbol{x}_{i-1})\boldsymbol{W}_k, \\
\boldsymbol{v}_t &= \mathrm{ddlerp}_v(\boldsymbol{x}_i, \boldsymbol{x}_{i-1})\boldsymbol{W}_v, \\
\boldsymbol{g}_t &= \mathrm{ddlerp}_g(\boldsymbol{x}_i, \boldsymbol{x}_{i-1})\boldsymbol{W}_g, \\
\boldsymbol{d}_t &= \mathrm{lora}_d(\mathrm{ddlerp}_d(\boldsymbol{x}_i, \boldsymbol{x}_{i-1})), \\
\boldsymbol{w}_t &= \exp(-\exp(\boldsymbol{d}_t)),
\end{aligned}
\tag{7}
$$

where $\boldsymbol{W}_{\{r,k,v,g\}} \in \mathbb{R}^{D \times D}, \boldsymbol{A}_w \in \mathbb{R}^{D \times D_w}, \boldsymbol{B}_w \in \mathbb{R}^{D_w \times D}$ are trainable weight matrices. By Equation 2 we have $\boldsymbol{y}_i \in \mathbb{R}^{D \times 1}$. The output of TM is given as

$$
\boldsymbol{o}_t = \mathrm{concat}\left(\mathrm{SiLU}(\boldsymbol{g}_t) \odot \mathrm{LayerNorm}(\boldsymbol{y}_t)\right)\boldsymbol{W}_o \in \mathbb{R}^{D \times 1}.
\tag{8}
$$

In CM, for inputs $\{\boldsymbol{x}'_i \in \mathbb{R}^D\}_{i=1}^T$, a linear interpolation (lerp) is first conducted as

$$
\mathrm{lerp}_{\{r',k',v'\}}(\boldsymbol{x}'_i, \boldsymbol{x}'_{i-1}) = \boldsymbol{x}'_i + (\boldsymbol{x}'_{i-1} - \boldsymbol{x}'_i)\boldsymbol{\mu}_{\{r',k',v'\}},
\tag{9}
$$

where $\boldsymbol{\mu}_{\{r',k',v'\}} \in \mathbb{R}^D$ is trainable parameter. CM is conducted as

$$
\begin{aligned}
\boldsymbol{r}'_t &= \mathrm{lerp}_{r'}(\boldsymbol{x}'_i, \boldsymbol{x}'_{i-1})\boldsymbol{W}_{r'} \in \mathbb{R}^D, \\
\boldsymbol{k}'_t &= \mathrm{lerp}_{k'}(\boldsymbol{x}'_i, \boldsymbol{x}'_{i-1})\boldsymbol{W}_{k'} \in \mathbb{R}^{D_{\mathrm{ffn}}}, \\
\boldsymbol{v}'_t &= \mathrm{ReLU}(\boldsymbol{k}'_t)^2 \boldsymbol{W}_{v'} \in \mathbb{R}^D, \\
\boldsymbol{o}'_t &= \sigma(\boldsymbol{r}'_t) \odot \boldsymbol{v}'_t \in \mathbb{R}^D,
\end{aligned}
\tag{10}
$$

where $\boldsymbol{W}_{r'} \in \mathbb{R}^{D \times D}, \boldsymbol{W}_{k'} \in \mathbb{R}^{D \times D_{\mathrm{ffn}}}, \boldsymbol{W}_{v'} \in \mathbb{R}^{D_{\mathrm{ffn}} \times D}$ are trainable matrices.

**MVHS layer**. The MVHS layer is modified from the TM layer in RWKV-6. For input sequence $\{\boldsymbol{x}_i \in \mathbb{R}^D\}_{i=1}^T$ of length $T$, MVHS layer first maps them into $\boldsymbol{k}_t, \boldsymbol{v}_t, \boldsymbol{w}_t$ as Equation 7. Then, given

number of heads $N$ and dimension of head $D_{\text{head}} = D/N$, the output $\boldsymbol{S}$ of the MVHS layer is given as

$$
\begin{aligned}
\boldsymbol{w}_i^h &= \boldsymbol{w}_i \left[hD_{\text{head}} : (h+1)D_{\text{head}} - 1\right] \in \mathbb{R}^{D_{\text{head}}}, \\
\boldsymbol{k}_i^h &= \boldsymbol{k}_i \left[hD_{\text{head}} : (h+1)D_{\text{head}} - 1\right] \in \mathbb{R}^{D_{\text{head}}}, \\
\boldsymbol{v}_i^h &= \boldsymbol{v}_i \left[hD_{\text{head}} : (h+1)D_{\text{head}} - 1\right] \in \mathbb{R}^{D_{\text{head}}}, \\
\boldsymbol{S}_i^h &= \sum_{t=1}^{i} \text{diag}\left(\bigotimes_{j=t+1}^{i} \boldsymbol{w}_j^h\right) \boldsymbol{k}_t^h (\boldsymbol{v}_t^h)^{\text{T}} \in \mathbb{R}^{D_{\text{head}} \times D_{\text{head}}}, h = 0, 1, \cdots, N-1, \\
\boldsymbol{S}_i &= \text{concat}(\{\boldsymbol{S}_i^h\}_{h=0}^{N-1}) \in \mathbb{R}^{N \times D_{\text{head}} \times D_{\text{head}}}.
\end{aligned}
\tag{11}
$$

This is equivalent to directly using the multi-channel MVHS of the TM in RWKV-6 as the output. The hidden state preserves the aggregated information of the input events, which is precisely what we want for the event feature. Moreover, the multi-channel matrix shape matches the multi-channel image-like handcrafted event representations and could convey expressive fine-grained spatial information. Therefore, we think using MVHS as output is a key towards expressive event features. The data flow of the EVA framework, from raw events to the targets, is summarized in Figure 6.

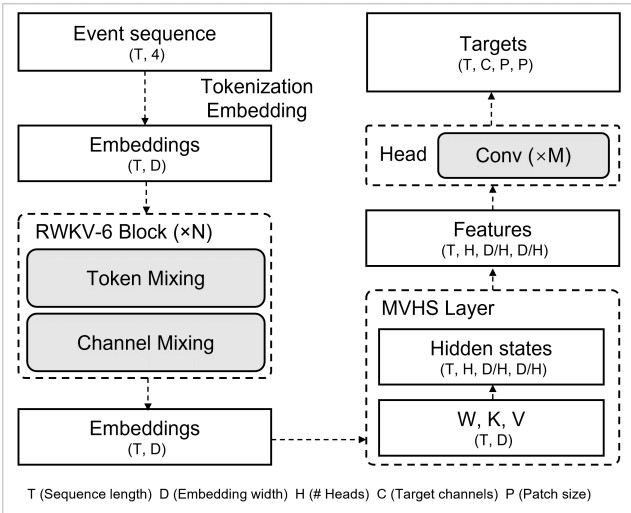

Figure 6: Data flow of the framework.

## C   DATASET PREPROCESS

**DVS128-Gesture**. DVS128-Gesture has 11 classes of human actions of 29 different users. The official split of the dataset uses users 0 to 23 as the train split and users 24 to 29 as the test split. We follow this method and use users 21 to 23 in the original train split as a validation set. The pretraining is conducted on the new training set (user 0 to 20) and validated on the val set (user 21 to 23). We report the test accuracy of the best model on the validation set.

For pretraining the model on sequences of length 8192, we first split all the files into patches of patch size $P = 16$. Then, we slice each patch into sequences of length 10240 with a stride of 8192 as samples. In each sample, the first 8192 events are the inputs to the model, and the last 2048 events are used to generate the NRP objectives. We save the samples in the data format of $\Delta t, x, y, p$ and convert them into uint16 to save storage. The difference of timestamps $\Delta t$ may be clamped to the numeric range of uint16, but this happens seldom and has little influence on the experiments.

**N-Cars**. N-Cars has files of duration 100 ms. Each file has around 4000 events. After splitting the files into patches of size $16 \times 16$, the average length of each patch is around 200, which is quite small. Therefore, on this dataset, we split the patches into samples of length 768 with a stride of 512. The first 512 events are inputs, and the last 256 events are used for NRP objectives. Following

(Gehrig et al., 2019; Cannici et al., 2020), we use 20% of the training set as the validation split. The samples are saved in the same format as DVS128-Gesture.

**Gen1**. We first filter out all the hot pixels caused by event camera sensor errors in the dataset. For every 10 ms in the file, the events in a pixel are deleted if they activate more than 40 times (we check the normal files and find that this value is no more than 10 regularly). This is similar to limiting the maximum value in the event counts in the synchronous method (Gehrig & Scaramuzza, 2023). We use the official train-val-test split of Gen1 and split the files into patches of size $16 \times 16$. Since Gen1 is a large dataset with sufficient events for pretraining the model, we do not use all the events for pretraining in one epoch. Instead, we uniformly sample a slice of length 4096 from each patch. The first 2048 events are inputs, and the last 2048 events are left for NRP objectives. For pretraining, we save the patches in the same format as DVS128-Gesture. For training, all the generated features are multiplied by a factor of 1/8, converted to their absolute value of data type uint8 to be consistent with the numerical range of the original inputs of RVT (0 to 20). We do this because we find that if we use the original float data type of the features, there will probably be a numerical NaN error caused by the YOLOX head (Ge et al., 2021). This could also help save storage.

## D   IMPLEMENTATIONS

Three asynchronous encoders are presented in the experiment. In the experiments on DVS128-Gesture, we present an encoder (EVA) with 3 RWKV-6 blocks and 1 MVHS layer with $D = 128, D_{\text{head}} = 8, N = 16, D_{\text{ffn}} = 256, D_{\text{lora}} = D_w = 16$. The shape of the MVHS output is $(16 \times 8 \times 8)$. To predict the representation of shape $(* \times 16 \times 16)$, we use a ResNet head with 1 basic ResNet block of width 64, 1 ConvTranspose2d$(in = 64, out = 32, ks = 4, stride = 2)$ to upsample the $8 \times 8$ hidden to $16 \times 16$ and 1 basic ResNet block of width 32. We use one $1 \times 1$ convolution layer as output. In the experiment on N-Cars, except for the encoder (EVA) on DVS128-Gesture, we also present an encoder with $N = 8, D_{\text{head}} = 16$ (referred to as EVA ($D_{\text{head}} = 16$) in the main context) since the spatial resolution of files in N-Cars is already cropped and small in spatial resolution. Using a shrunk hidden state as feature could indeed help reduce downstream tasks inference latency, but will decrease the accuracy (see Table 2). To keep the same total size of the feature, we use only half the channels in the MVHS layer, that is $N = 4, D_{\text{head}} = 16$. The shape of the MVHS output is $4 \times 16 \times 16$. In this case, since there is no need for upsampling, we use a ResNet with 1 basic ResNet block of width 32 as the head. In the experiment on Gen1, we also present a larger encoder with $D = 192, D_{\text{head}} = 16, N = 8, D_{\text{ffn}} = 384$ (referred to as EVA-L ($D = 192, D_{\text{head}} = 16$) in the main context) in the RWKV-6 blocks and $N = 6, D_{\text{head}} = 16$ in the MVHS layer. The same ResNet head as N-Cars is used for pretraining.

For initialization, we use the official initialization method in RWKV-6 for the encoder and Kaiming initialization for the ResNet. The asynchronous encoder, including the RWKV-6 blocks and MVHS layer, is trained with bfloat16 precision as the default value in the original implementations in RWKV-6.

In the recognition task on DVS128-Gesture, we use a 14-layer (ResNet-14) as the classifier by removing the last stage of the original ResNet-18 (He et al., 2016) to reduce inference latency (see Table 12). Interestingly, ResNet-14 has a higher accuracy on the validation set with a smaller parameter size and lower latency. We think this is because a smaller model could help overcome overfitting. Despite this, we find that ResNet-18 could also outperform previous work (Martin-Turrero et al., 2024). We use batch size 32, Adam optimizer with learning rate 0.001, mixup augmentation, and train 200 epochs on a single RTX 3090 GPU with a decay of learning rate $\gamma = 0.8$ every 20 epochs. In the experiments on N-Cars, we follow (Cannici et al., 2020; Gehrig et al., 2019) to use a

Table 12: Classification results on DVS128-Gesture with ResNet-18

| Model | Classifier | # Params | Latency | Val. acc. (SA) | Test acc. (SA) |
|-------|-----------|----------|---------|----------------|----------------|
| ALERT-Tr. | RM | 13.96 M | 9.6 ms | – | 84.6% |
| EVA | ResNet-18 | 11.28 M | 2.0 ms | 92.8% | 92.0% |
| EVA | ResNet-14 (used) | 2.83 M | 1.5 ms | 94.1% | 92.9% |

ResNet-34 pretrained on ImageNet as the classifier. Following the train settings of previous work,

we use a batch size of 64, Adam optimizer with a constant learning rate of 0.0001, random flip as data augmentation, and train 100 epochs on an RTX 3090 GPU. In the experiments on Gen1, we use the default training setups of RVT(Gehrig & Scaramuzza, 2023) for a fair comparison. We use batch size 8, learning rate 2e-4, and train 400,000 steps on a single A800 GPU. The training takes about 2 days.

## E  PRETRAINING OBJECTIVES

We use event count (EC) and time surface (TS) as the pretraining objectives. Consider events $\mathcal{E} = \{e_i = (t_i, x_i, y_i, p_i)\}_{i=1}^T$ and patch size $P$, EC of a given time window $[t_s, t_e]$ is constructed as a 2 channel image of shape $2 \times P \times P$ by

$$\text{EC}(p, x, y) = \sum_i 1_{x_i=x, y_i=y, p_i=p, t_i \in [t_s, t_e]} \in \mathbb{R}^{2 \times P \times P}. \tag{12}$$

and TS is calculated as

$$\text{TS}(p, x, y) = \max_i \exp\left(\frac{t_i - t_T}{\tau}\right) \times 1_{x_i=x, y_i=y, p_i=p} \in \mathbb{R}^{2 \times P \times P}, \tag{13}$$

where $\tau > 0$ is a time constant.

Table 13: Objectives for the SSL. For EC in MRP

| Dataset | MRP | | NRP |
| --- | --- | --- | --- |
| | EC time window (ms) | TS $\tau$ (ms) | EC time window (ms) |
| DVS128-Gesture | 100 | 100 | 20 |
| N-Cars | 100 | 100 | 20 |
| Gen1 | 50, 25, 10, 5 | 200 | 10 |

Calculating the loss of the predictions and targets at every individual event would demand prohibitive GPU memory. To mitigate this, we divide the event sequence into chunks, and calculate the loss only at the end of each chunk. For DVS128-Gesture dataset, we use chunk length $T_{\text{chunk}} = 512$. For N-Cars and Gen1, we use chunk length $T_{\text{chunk}} = 16$ .

The objectives in the experiments are given in Table 13. On Gen1, we use multiple ECs and larger TS $\tau$ because we find it could help converge at the initial period of pretraining to decrease the time required to pretrain on the large dataset.

Table 14: Effect of SSL objectives on DVS128-Gesture

| MRP | | NRP | MRP Loss | | NRP Loss | Acc. | |
| --- | --- | --- | --- | --- | --- | --- | --- |
| EC (ms) | TS (ms) | EC (ms) | EC | TS | EC | FVA | SA |
| 100 | 100 | 20 | 0.366 | $3.16 \times 10^{-3}$ | 0.767 | 98.1% | 94.7% |
| $100 \to 50$ | 100 | 20 | 0.200 | $2.33 \times 10^{-3}$ | 0.764 | 95.5% | 94.3% |
| $100 \to 200$ | 100 | 20 | 0.807 | $3.60 \times 10^{-3}$ | 0.767 | 95.5% | 94.3% |
| 100 | $100 \to 50$ | 20 | 0.398 | $2.39 \times 10^{-3}$ | 0.766 | 98.1% | 94.1% |
| 100 | $100 \to 200$ | 20 | 0.374 | $3.78 \times 10^{-3}$ | 0.780 | 96.2% | 94.4% |
| 100 | 100 | $20 \to 10$ | 0.389 | $3.29 \times 10^{-3}$ | 0.364 | 97.4% | 94.2% |
| 100 | 100 | $20 \to 40$ | 0.353 | $2.86 \times 10^{-3}$ | 1.698 | 98.7% | 94.3% |

For a better understanding of the effects of the SSL objectives, here we give a study on DVS128-Gesture. The results are given in Table 14. We change the SSL objectives and find that the validation set accuracy (SA) is not heavily affected. The optimal configuration of the hyperparameter has not been investigated yet, and we hope to leave it for future study.

## F    VISUALIZATION

We visualize the handcrafted representations for SSL pretraining, the predicted results of the pre-training target, and the learned feature in Figure 7. The representations and results are generated on DVS128-Gesture.

Figure 7 (a) provides the visualization of handcrafted representations (targets in SSL), and Figure 7 (b) shows the visualization of the reconstructed representations (predictions in SSL). The comparison between Figure 7 (a) and (b) demonstrates that during SSL, our neural network heads could consistently predict the corresponding handcrafted representations (EC, TS) with high similarity from the learned feature map. This shows that the learned features contain adequate information to reconstruct the handcrafted representations, and further shows that the EVA encoder is able to capture the majority of the spatial information from raw individual events. Additionally, the predicted EC (NRP) shows favorable consistency with its ground truth targets, demonstrating the success of predicting future values from the learned feature map. This indicates that our model is capable of learning short-term motion patterns rather than merely memorizing the inputs.

Figure 7 (c) provides the visualization of the learned feature map. Each figure is one channel of the MVHS layer output, and is a concatenation of multiple patches. Qualitatively, several channels of the learned feature map show patterns similar to the handcrafted representations, but with different windows of temporal accumulation (several channels preserve more past events, while others only maintain recent events). Besides, some channels show different patterns compared with the handcrafted representations. We leave a comprehensive quantitative analysis of the dynamics of the learned feature maps for further study.

## G    DETAILS OF THE MVHS AND PATCH-WISE ENCODING TO REDUCE MODEL SIZE

Assume that we keep the size of the learned feature unchanged. For original vector-value output, the output size is $D$, and the model parameter size is around $\mathcal{O}(D^2)$ (since the majority of the parameters come from the matrix). For MVHS output, the output size is $N \times (D'/N) \times (D'/N)$ with model size $\mathcal{O}(D'^2)$. Let $D = N \times (D'/N) \times (D'/N)$, we have $\mathcal{O}(D^2)/\mathcal{O}(D'^2) = D/N = D_{\text{head}}$. In our EVA, this factor is 8 or 16.

For patch-wise encoding, assume that we learn a feature that is proportional to the size of the objective representations. Consider an event camera with resolution $(H, W)$ (assume $H = W$ for simplicity) and patch size $P$, the number of patches $\# \; patches$ is $H^2/P^2$. Given the shape of the learned feature as $(N \times D \times D)$, if we use it to predict the objective representations of full size, which is $(* \times H \times H)$, then we have $D = H$ and the model size is around $\mathcal{O}(H^2)$. However, if we use the learned feature to predict the representations of a patch, which has the shape of $(* \times P \times P)$, we have $D = P$, and the model size is around $\mathcal{O}(P^2)$. Therefore, the model size decreases $\mathcal{O}(H^2)/\mathcal{O}(P^2) = \# \; patches$ times. In the experiment on DVS128-Gesture, we have $H = W = 128$ and $P = 16$. This factor is 64. On Gen1 with $H = 240, W = 304$ and $P = 16$, we have $\# \; patches = 285$.

## H    LIMITATIONS

Our work is limited in two aspects. First, while our EVA framework is evaluated on publicly available datasets recorded in real-world settings, especially in automotive driving scenarios, a full exploration of its utility in practical applications requires implementing the models on device-side, such as on FPGAs, and testing the whole framework in actual deployment scenarios. These aspects are not involved in the current study. Secondly, the models trained are relatively small in parameter size compared to pretrained large language models (LLMs) with billions of parameters. Consequently, the potential of self-supervised pretraining could not be fully exploited in our study. The parameter size of the asynchronous encoder is constrained by the requirement for real-time processing of high-temporal-resolution events. We aim to address this limitation in the future by decreasing the per-event computational complexity, thereby enabling the design of larger pretrained models capable of asynchronously processing events in real time.

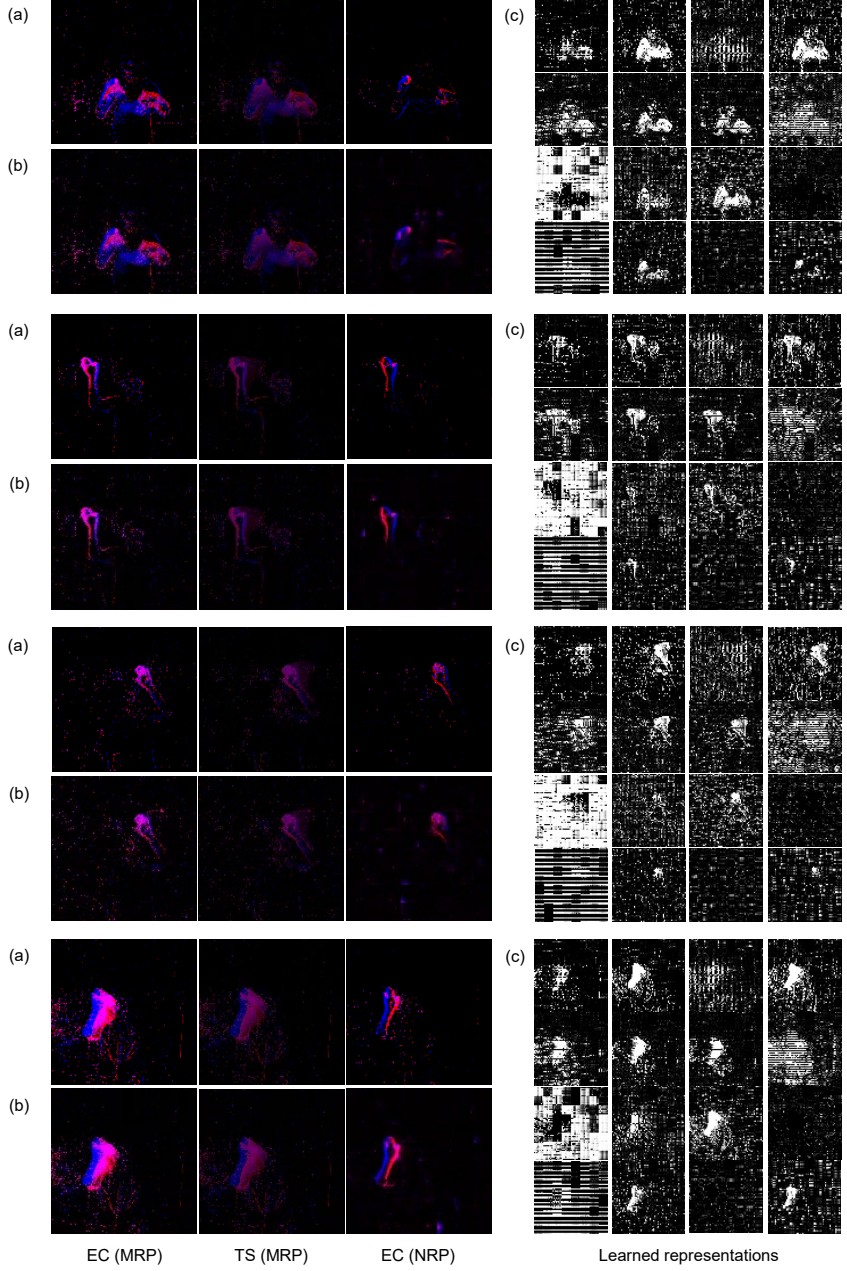

EC (MRP)          TS (MRP)          EC (NRP)          Learned representations

Figure 7: Visualization of 4 different gestures on DVS128-Gesture. (a) Ground truth of the hand-crafted representations. From left to right: EC, TS, and EC in the future. (b) Neural network head predicted handcrafted representations in the pretraining stage. (c) The learned features. Each figure is one channel of the MVHS layer output.

## I   COMPLEXITY ANALYSIS

Each update of the EVA features requires $\mathcal{O}(M)$ computations, where $M$ is the parameter size of the EVA encoder. Therefore, after a sequential length of T, the total MACs is proportional to the input length $T$ and the model size $M$ as $\mathcal{O}(TM)$. Here we summarize the complexity of other event-representing methods in Table 15. For EST, each time a new event arrives, the representations should be recomputed. The complexity is proportional to the number of events involved in the model size $M$, time window $L$, and number of channels of the EST representation $C$. For MatrixLSTM with temporal bins, the complexity is $\mathcal{O}(LM)$, which serves as a synchronous representation. For the Voxel grid, the complexity is $\mathcal{O}(L+HWC)$, and for ERGO, the complexity is $\mathcal{O}(LC+HWC)$.

Table 15: Complexity and estimated FLOPs of different representations. We consider a $L = 1 \times 10^5, H = 720, W = 1280$ event sequence on the 1Mpx dataset.

| Method | Asynchronous | Complexity per Event | Estimated FLOPs per Event |
|---|---|---|---|
| EVA-L | ✓ | $\mathcal{O}(M)$ | 1.4 M |
| ALERT-Tr. (RM) | ✓ | $\mathcal{O}(M)$ | 1.2 M |
| EST | | $\mathcal{O}(LMC + HWC)$ | 1152 M |
| MatrixLSTM | | $\mathcal{O}(LMC)$ | $> 100$ M |
| Voxel grid | | $\mathcal{O}(L + HWC)$ | 11 M |
| ERGO-12 | | $\mathcal{O}(LC + HWC)$ | 11 M |

## J   LLM USAGE

Large language models (LLMs) are used to improve the writing of the manuscript, including translating languages, correcting grammatical and spelling errors, and improving sentence clarity.

