# OpenReview forum: "Maximizing Asynchronicity in Event-based Neural Networks"
_ICLR.cc/2026/Conference — ICLR 2026 Poster_

### Official Review · Reviewer_A3qJ · 2025-10-27

**Soundness:** 3
**Presentation:** 3
**Contribution:** 3
**Rating:** 6
**Confidence:** 3

**Summary:**

This paper addresses the sparsity and asynchrony of event camera data by proposing an Asynchronous-to-Synchronous (A2S) framework to better align event streams with deep learning methods. Inspired by language models, the authors treat event streams as ordered and incrementally evolving information, introducing Event Asynchronous Feature Learning (EVA), which leverages linear attention and self-supervised learning. Experimental results demonstrate that the proposed method achieves strong performance on both event-based recognition and inspection tasks.

**Strengths:**

1. The motivation is clear. This paper specifically addresses the sparsity and asynchrony of event camera data and proposes a targeted solution. Treating continuous events as ordered natural language sequences and tokenizing them is a novel idea.

2. The proposed method plays a positive role in bridging event camera data with deep learning approaches, effectively promoting progress in this research area.

3. The writing is fluent overall, with a clear and coherent logical structure.

**Weaknesses:**

1. Insufficient experimental datasets. For the object recognition task, the authors only used relatively simple datasets such as N-Cars and DVS128-Gesture, without evaluating the method on more complex datasets like N-Caltech101 and N-ImageNet. This limitation may reduce the persuasiveness of the proposed approach.

2. Low resolution of experimental datasets. The datasets used—N-Cars, DVS128-Gesture, and Gen1—all have relatively low resolutions, whereas modern event cameras can already reach 1280×720. Therefore, it is highly necessary to test the proposed method on high-resolution datasets. Moreover, during the A2S conversion, the resolution of the event camera can significantly influence the transformation speed. The authors should discuss how resolution affects the performance and efficiency of their method.

**Questions:**

1. From Figure 4, it appears that MRP and NRP are not directly used in downstream tasks, but rather serve to guide the training of the event embedding module?

2. In Figure 3C, the event embedding is transformed by the task head into a representation, which is then used to compute the loss together with features derived from MRP and NRP. How are MRP and NRP themselves trained in this setup?

3. During the self-supervised learning process, how are MRP and NRP trained using handcrafted features? Are they trained separately in advance? Does this pretraining need to be done for each task or dataset individually? Why not directly use handcrafted features to train the event embedding instead?

4. In Figure 6, what exactly are the learned representations, and what insights do they provide?

5. Is the representation generalizable to all event-based downstream tasks, including optical flow estimation, semantic segmentation, etc., or is it restricted to specific tasks only?

6. Compared with directly stacking event data into event images and processing them, what advantages does the proposed method offer? Does it provide benefits in terms of inference speed or computational cost (FLOPS)?

---

> ### Author Response · Authors · 2025-11-26
> **Reply to Weakness 1**
>
> Dear reviewer A3qJ,
>
> Thank you for your helpful suggestions and review. Here are our replies.
>
> ### Weaknesses:
> > 1.	Insufficient experimental datasets. For the object recognition task, the authors only used relatively simple datasets such as N-Cars and DVS128-Gesture, without evaluating the method on more complex datasets like N-Caltech101 and N-ImageNet. This limitation may reduce the persuasiveness of the proposed approach.
>
> Thank you for your helpful comments. We add new experimental results on the N-Caltech101 dataset as you suggested, and the results are shown as follows in Table 1.
> From the experimental results, our EVA method outperforms all the asynchronous methods (10%+), and has competitive performance compared with synchronous methods, only 0.3% less than E2VID, which reconstructs the gray-scale image and introduces additional latency.
> Moreover, our method surpasses synchronous methods, including EST, Pillar, and Matrix-LSTM. Note that our method does not introduce an additional representing latency, while synchronous methods have to wait for a fixed update time. This could demonstrate the expressivity and efficiency of our method.
>
> Since large-scale event-based classification datasets, such as N-ImageNet and ES-Image, are converted by algorithms (Not by real event cameras). Therefore, the events in these datasets share identity timestamps, and thus are NOT asynchronous in principle. Therefore, our method does not apply to these simulated datasets, because we demand asynchronous inputs. We want to emphasise that this does not affect the contribution of our method, since our method focuses on a real-time application scenario for low-latency perception tasks with real event cameras.
>
> #### Table 1. Recognition results on the N-Caltech101 dataset. A. denotes asynchronous methods, S. denotes synchronous methods.
> | Model | Type | Acc. (%) |
> |-------|------|------|
> | HOTS      | A. | 21.0 |
> | NVS-S     | A. | 67.0 |
> | AEGNN     | A. | 66.8 |
> | Asynet    | A. | 74.5 |
> | FARSE-CNN | A. | 68.7 |
> | EST       | S. | 81.7 |
> | Matrix-LSTM      |  S.  | 84.3 |
> | Pillar           |  S.  | 85.3 |
> | E2VID            |  S.  | **86.6** |
> | EVA-L (ours)     |  A2S | $\underline{86.3}$ |
>
>
> To better demonstrate the expressivity of our EVA method, we also add extended experiments on Gen1 as suggested by Reviewer n32k. We fix the backbone as Swin Transformer V2, and compare event representations including Time-Surface (HOTS), Voxel-Grid, TORE, ERGO, and Pillar. The results are shown in Table 2.
> Our EVA method obtains the best accuracy among asynchronous methods,
> For synchronous methods, our EVA method also achieves competitive results, while maintaining extremely low representing latency (0.002 ms).
> We hope these results could further validate the effectiveness of our EVA framework.
>
> #### Table 2. Detection results on the Gen1 dataset with an identical backbone.
>
> | Method | mAP | Representing latency (ms) |
> | :--- | :---: | :---: |
> | Voxel (bin=12) | 39.5 | 16.3 |
> | ERGO-12 (w aug.)  | 50.4 | > 10 |
> | ERGO-12 (w/o aug.)  | 49.3 | > 10 |
> | Pillar| 53.1 | 11.4 |
> | HOTS | 49.0 | 1.0 |
> | TORE | 43.6 | < 0.1 |
> | EVA-L (ours)| 49.7 | < 0.1 |

---

> ### Author Response · Authors · 2025-11-26
> **Reply to Weakness 2**
>
> > 2.	Low resolution of experimental datasets. The datasets used—N-Cars, DVS128-Gesture, and Gen1—all have relatively low resolutions, whereas modern event cameras can already reach 1280×720. Therefore, it is highly necessary to test the proposed method on high-resolution datasets. Moreover, during the A2S conversion, the resolution of the event camera can significantly influence the transformation speed. The authors should discuss how resolution affects the performance and efficiency of their method.
>
> Thank you for your advice. It is quite true that modern event cameras have high resolutions of 1280×720, while our experiments are conducted on Gen1 with a resolution of 240×304. This is a common limitation of current asynchronous event processing methods, since higher resolution leads to larger sequential length, while for event-by-event processing, the computation cost increases linearly with the sequential length.
> Current SOTA asynchronous methods methods (DAG-r[1], FARSE-CNN[2], ALERT-Tr.[3]), are all evaluated on low-resolution datasets (Gen1, DSEC-Dection,DVS128-Gesture, etc.). We also tried to train on 1Mpx dataset (1280x720), but limited by computation resources, it takes us more than 60 days to conduct a complete training with 8 3090 GPUs. Therefore, we are sorry that we can currently not provide the results. We hope to explore training larger models on such a large-scale dataset if we could get more computational resources in the future.
>
> To discuss the influence of resolution on the performance and efficiency of our method, we provide the following analysis.
>
> (1) Theoretically, the performance of our method would not be affected by the resolution. Since we conduct a patch-wise encoding (PWE) strategy, our EVA model is trained on non-overlapping patches (16x16). No matter what the sensor resolution is, each patch is encoded independently. Therefore, the performance of the asynchronous encoder would not be affected by the resolution.
>
> (2) In principle, the latency of our method is NOT limited by the resolution.
> Since we conduct a PWE strategy, all the patches could be computed in parallel. Therefore, the latency is only determined by the longest computation time among all the patches, and is determined by the Patch Event Rate (number of events per second in each patch). This item does not change significantly with resolution, as presented in Table 3. Therefore, our method could theoretically apply to event cameras at any resolution.
>
> ####  Table 3. Event Rates of different datasets. (Event Rate: number of events per second; Patch Event Rate: number of events per second in each patch)
> | Dataset          | DVS128-Gesture | N-Cars | Gen1  |
> |------------------|----------------|--------|-------|
> | Resolution       | 128x128        | 100x120| 240x304|
> | Avg. Event Rate (K/s) | 58             | 39     | 618   |
> | Patch Size       | 16x16          | 16x16  | 16x16 |
> | Patch Event Rate (K/s)| 0.9   | 4.1 | 2.2 |
>
> (3) Practically, however, we admit that the processing speed is limited by the parallel computing cores of the hardware. And your comment is really critical for the asynchronous event processing area. We want to claim that this is not the problem of our method solely, but the common open problem of asynchronous event-based vision. Existing methods use subsampling, temporal pooling or spatial-temporal aggregation to reduce event count, but lose performance (both in accuracy and latency).
>
> (4) The central aim of asynchronous event processing is to keep event-by-event processing capability (thus low latency), while promoting the model performance (towards dense synchronous methods), or keep the performance but reduce the event-wise computation (thus could be leveraged to large resolution).
> Compared with our A2S baseline (ALERT Transformer), which is only used on recognition tasks (DVS128, 128x128, NCars, 100x120), our method has already made a step forward by successfully applying to detection tasks (Gen1) with the same MACs but a large improvement in performance. Therefore, we think that this open problem in the area does not affect the contribution of our method.
>
> Since this is a common interest problem in the asynchronous event processing area. We are looking forward to your further comments and insight about this problem. Thank you!
>
>
> [1] Gehrig, Daniel, and Davide Scaramuzza. "Low-latency automotive vision with event cameras." Nature 629.8014 (2024): 1034-1040.
>
> [2] Santambrogio, Riccardo, Marco Cannici, and Matteo Matteucci. "Farse-cnn: Fully asynchronous, recurrent and sparse event-based cnn." European Conference on Computer Vision, 2024.
>
> [3] Martin-Turrero, Carmen, et al. "Alert-transformer: Bridging asynchronous and synchronous machine learning for real-time event-based spatio-temporal data." arXiv preprint arXiv:2402.01393 (2024).

---

> ### Author Response · Authors · 2025-11-26
> **Reply to Questions 1~3**
>
> ### Questions:
>
> > 1.	From Figure 4, it appears that MRP and NRP are not directly used in downstream tasks, but rather serve to guide the training of the event embedding module?
>
> Yes. We use handcrafted event representations as the training targets. While we do not need them in downstream tasks.
>
> The main reason for this is that (1) these handcrafted representations could be synchronous, and thus will not be able to be sampled in demand for downstream tasks. (2) The NRP task requires future events, which are not available in real-time downstream tasks.
>
> > 2.	In Figure 3C, the event embedding is transformed by the task head into a representation, which is then used to compute the loss together with features derived from MRP and NRP. How are MRP and NRP themselves trained in this setup?
>
> Thanks for your question. Here we want to clarify that MRP and NRP in Figure 3C do not refer to a neural network module, but refer to an SSL task. MRP means training the EVA encoder by predicting handcrafted representations, and NRP means training the EVA encoder by predicting future events. These targets are directly calculated from the raw event stream, without any neural network module.
>
> Since Figure 3C could be misleading, we made corresponding changes to the Figure in the revised manuscript. We sincerely thank you for pointing out this problem.
>
> > 3.	During the self-supervised learning process, how are MRP and NRP trained using handcrafted features? Are they trained separately in advance? Does this pretraining need to be done for each task or dataset individually? Why not directly use handcrafted features to train the event embedding instead?
>
> Thank you for your question. As question 3 explained, MRP and NRP refer to tasks instead of a neural network module. We are exactly using the handcrafted event representations to train the event embedding, just as you described.
>
> We are sorry for the misleading Figure 3C in the manuscript, and we have correspondingly refined it. Please just point out if there are any problems that you think should be clarified regarding this.

---

> ### Author Response · Authors · 2025-11-26
> **Reply to Questions 4~6**
>
> ### Questions
>
> > 4.	In Figure 6, what exactly are the learned representations, and what insights do they provide?
>
> Thank you for your question. Figure 6C shows the learned representations, with each picture standing as 1 channel of the feature map. Here we provide several observations from the visualization results:
>
> (1) It is shown that the predicted EC and TS in the MRP task (Fig. 6 (b)) are quite the same as their targets. This could demonstrate that the EVA model could effectively learn to capture the majority of spatial information from just the raw event inputs.
>
> (2) From the similarity of the predicted EC in the NRP task (Fig. 6 (a)) with their targets, we could verify that the EVA model could effectively predict the future, but not simply copy the current input.
>
> (3)  We find that several channels of the learned feature show patterns similar to the handcrafted representations, but with different windows of temporal accumulation. These could indicate that the model is learning to capture temporal dynamics in a more flexible manner than fixed handcrafted representations. This could also be observed in other optimized event representations like ERGO, Figure 9 in supplementary material of [4]
>
> [4] Zubić, Nikola, et al. "From chaos comes order: Ordering event representations for object recognition and detection." Proceedings of the IEEE/CVF International Conference on Computer Vision. 2023.
>
> > 5.	Is the representation generalizable to all event-based downstream tasks, including optical flow estimation, semantic segmentation, etc., or is it restricted to specific tasks only?
>
> Thanks for your question. Theoretically, the learned asynchronous feature could generalize to all the event-based downstream tasks, since it could predict EC and TS, and EC and TS are applicable to the majority of the event-based downstream tasks.
>
> To further support this claim, we add further experiments on Optical Flow estimation tasks. Due to the limitation of computational resources, the experiments are currently not finished. We will update the results immediately as they are available before the Discussion Phase.
>
>
> > 6.	Compared with directly stacking event data into event images and processing them, what advantages does the proposed method offer? Does it provide benefits in terms of inference speed or computational cost (FLOPS)?
>
> Thanks for your question. Compared with converting events into images directly, our method has advantages in lower latency and better performance.
>
> (1) For latency, directly stacking event data into event images is, in principle, synchronous. They require an additional accumulation/computation time. Accumulation time means you have to wait for a fixed time window to gather enough events, and then stack them into images. Computation means you have to process all the events in the time window to generate the event images. Both of these introduce extra latency.
>
> (2) For performance, directly stacking events through the temporal dimension loses the temporal information of events and cannot leverage the fine-grained asynchronous information. Therefore, many event representing methods, like EST, ERGO, are proposed to improve the stacking paradigm with learned kernel, so that they could better preserve the spatio-temporal information from raw events. However, although these methods could achieve SOTA accuracy with dense neural networks, they add additional computation and latency. Our method has both advantages. It not only reduces the latency by a recurrent updating formula, but reserves spatial-temporal information by its asynchronous inputs.
>
> Here we will conduct more experiments to support our claims. We compare our EVA method with stacking-based representation methods, including EC (fixed time window), EC (fixed event number), and Voxel grid on the Gen1 dataset with an identical backbone (Swin Transformer V2). The results are shown in Table 4. It could be seen that our EVA method outperforms all stacking-based representing methods, which could further validate the expressivity of our method.
>
> ####  Table 4. Comparison with stacking-based representing methods on the Gen1 dataset with an identical backbone.
>
> | Backbone | Swin v2 | RVT |
> | :--- | :---: | :---: |
> | EC (Fixed time window)  | 43.9 | 46.9 |
> | EC (Fixed event number) | 44.0 | 46.2 |
> | Voxel grid | 39.5   |  47.2  |
> | EVA-L (ours)| **49.7** | **47.7** |
> | EVA (ours) | 49.0 | 47.5  |

---

> > ### Author Response · Authors · 2025-12-03
> > **Reply to Question 5**
> >
> > > 5. Is the representation generalizable to all event-based downstream tasks, including optical flow estimation, semantic segmentation, etc., or is it restricted to specific tasks only?
> >
> > Here we also evaluate our method on optical flow estimation on the MVSEC dataset and follow the experimental settings of EV-FlowNet. We compare average endpoint error (AEE). As shown in Table 5, our method is also applicable to optical flow estimation.  Our method outperforms the asynchronous method HUGNet-2, while giving similar results to the synchronous method EV-FlowNet but without representing latency, which further demonstrates the effectiveness of our method.
> >
> > #### Table 5: Optical flow estimation results on the MVSEC dataset.
> >
> > | Method                                   | Type | AEE |
> > |------------------------------------------|------|-----|
> > | HUGNet-2 (Dampfhoffer et al., 2025)      | A.   | 1.52 |
> > | EV-FlowNet (Zhu et al., 2018)            | S.   | 0.96 |
> > | Spike-FlowNet (Lee et al., 2020)         | S.   | **0.79** |
> > | EVA-L (ours)                             | A2S  | 1.01 |

---

### Official Review · Reviewer_n32k · 2025-10-30

**Soundness:** 4
**Presentation:** 3
**Contribution:** 3
**Rating:** 6
**Confidence:** 5

**Summary:**

This paper presents EVA, a novel asynchronous-to-synchronous (A2S) processing framework for event cameras. EVA builds an asynchronous encoder based on a linear attention mechanism and leverages self-supervised learning to learn generalizable event representations. It significantly outperforms existing A2S methods across multiple event-based vision tasks, including object recognition and object detection. Notably, EVA achieves 47.7 mAP on the Gen1 detection dataset, and demonstrating its strong representation capability and generalization across complex tasks.

**Strengths:**

1. The paper draws an insightful analogy between event streams and language sequences, introducing a linear attention mechanism and self-supervised learning to propose a novel and efficient paradigm for asynchronous event processing.

2. The authors conduct extensive evaluations on multiple public datasets (DVS128-Gesture, N-Cars, and Gen1), covering both recognition and detection tasks. The results are convincing and demonstrate clear advantages over various baseline methods.

3. The manuscript is well written and logically structured, with detailed figures and tables, and a comprehensive appendix that further supports the work.

**Weaknesses:**

1. As an A2S framework, a key question is how to achieve better asynchronous event representation. While I acknowledge that the authors have already included sufficient baseline comparisons, I wonder whether they have considered fixing the task-specific backbone and varying only the event representations, as described in Section 2.1. For example, in the object detection task on Gen1, one could fix a backbone (e.g., RVT-B) and compare different existing representation methods (e.g., time-surface, Matrix-LSTM, EST, voxel grid, graph-based). You may refer to Table 1 in ACGR [1] for a relevant comparison format.

2. Although Appendix H provides latency analysis, it focuses only on comparisons within EVA itself. It would be more informative to include approximate inference time and computational complexity comparisons across different open-source representation methods (e.g., Matrix-LSTM, EST, voxel grid, graph-based). Since event representation is a fundamental technique, demonstrating its low-latency characteristics is particularly important.

3. In future journal extensions, the authors could consider including more visualization results in the main text to better highlight the advantages of the proposed method.

[1] Asynchronous Collaborative Graph Representation for Frames and Events, CVPR 2025.

**Questions:**

1. The fitting targets during self-supervised training in this paper are EC and TS. Ideally, after training is completed, the proposed method could become some kind of weighted mixture of EC and TS. So why not directly use EC and TS? For example, in the experiments presented in this paper, what would be the performance change if the input representation were replaced with EC and TS?

2. Integration of events and RGB frames is an emerging trend. Could this framework be extended in the future to design a joint asynchronous representation that fuses both modalities? It would be interesting to hear the authors’ thoughts or potential directions on this.

3. Introducing NLP-inspired approaches into event representation is a novel idea. What specific advantages does this bring compared to conventional designs? Could the authors provide a more theoretical explanation or justification for this perspective?

4. Looking ahead, could this asynchronous representation technique be incorporated into event-based multimodal large language models [2]? This would be an exciting direction for exploring the intersection between event-based vision and multimodal reasoning.

[2] EventGPT: Event Stream Understanding with Multimodal Large Language Models, CVPR 2025.

---

> ### Author Response · Authors · 2025-11-26
> **Reply to Weakness**
>
> Dear reviewer n32k,
>
> Thank you for your careful and helpful review. Here are our replies:
>
> ### Weaknesses:
>
> > 1.	As an A2S framework, a key question is how to achieve better asynchronous event representation. While I acknowledge that the authors have already included sufficient baseline comparisons, I wonder whether they have considered fixing the task-specific backbone and varying only the event representations, as described in Section 2.1. For example, in the object detection task on Gen1, one could fix a backbone (e.g., RVT-B) and compare different existing representation methods (e.g., time-surface, Matrix-LSTM, EST, voxel grid, graph-based). You may refer to Table 1 in ACGR [1] for a relevant comparison format.
>
> Thank you for your valuable advice and guidance. Fixing the backbone and comparing the existing representation methods could indeed better demonstrate the expressivity of the EVA method. Therefore, we add extended experiments on Gen1 as suggested. We fix the backbone as Swin Transformer V2, and compare event representations including Time-Surface (HOTS), Voxel-Grid, TORE, ERGO, and Pillar. The results are shown in Table 1.
> Our method obtains the best accuracy among asynchronous methods,
> For synchronous methods, our EVA method also achieves competitive results, even surpassing ERGO-12 (w/o aug.), while maintaining extremely low representing latency (0.002 ms). These results could demonstrate the expressivity of EVA.
>
> Due to time limitations, we do not conduct experiments on Matrix-LSTM and EST, because their time consumption is extremely high with the Gen1 dataset. In Table 2, we provide recognition results on N-Caltech101  and N-Cars datasets, with an identical ResNet-34 backbone pretrained on ImageNet. Our EVA outperforms both EST and Matrix-LSTM, which could further validate the expressivity of our method.
>
> The results are also added in the revised manuscript in Appendix A.
>
> #### Table 1. Detection results on the Gen1 dataset with an identical backbone Swin v2.
>
> | Method | mAP | Representing latency (ms) |
> | :--- | :---: | :---: |
> | Voxel (bin=12) | 39.5 | 16.3 |
> | ERGO-12 (w aug.)  | 50.4 | > 10 |
> | ERGO-12 (w/o aug.)  | 49.3 | > 10 |
> | Pillar| 53.1 | 11.4 |
> | HOTS | 49.0 | 1.0 |
> | TORE | 43.6 | < 0.1 |
> | EVA-L (ours)| 49.7 | < 0.1 |
>
> #### Table 2. Recognition results with an identical backbone ResNet-34.
> | Model | N-Cars | N-Caltech101 |
> |-------|------|------|
> | EST              | 92.5 | 83.7  |
> | Matrix-LSTM      |  95.8  | 83.5 |
> | EVA-L (ours)     |  **96.3** | **86.3** |
>
>
> > 2.	Although Appendix H provides latency analysis, it focuses only on comparisons within EVA itself. It would be more informative to include approximate inference time and computational complexity comparisons across different open-source representation methods (e.g., Matrix-LSTM, EST, voxel grid, graph-based). Since event representation is a fundamental technique, demonstrating its low-latency characteristics is particularly important.
>
> Thank you for your helpful advice. To better demonstrate the latency advantage of our method, and include an analysis of different representation methods in Appendix I. We summarize the results here in Table 3.
>
>
> #### Table 3. Latency and computational complexity comparisons of different representation methods on the Gen1 dataset.
>
> | Method | Asynchronous | Complexity per Event | Estimated FLOPs per Event |
> | :--- | :---: | :--- | :--- |
> | EVA-L | ✓ |  $\mathcal{O}(M)$  | 1.4 M |
> | ALERT-Tr. (RM) | ✓ | $\mathcal{O}(M)$ | 1.2 M |
> | EST | | $\mathcal{O}(LMC + HWC)$ | 1152 M |
> | MatrixLSTM | | $\mathcal{O}(LMC)$ | $> 100$ M |
> | Voxel grid | | $\mathcal{O}(L+HWC)$ | 11 M |
> | ERGO-12 | | $\mathcal{O}(LC+HWC)$ | 11 M |
>
>
> Here, M is the model parameter size, L is the number of events, H and W are the height and width of the sensor, and C is the number of channels in the representation.
>
> The estimated FLOPs are calculated with the following settings: L=$10^5$, H=720, W=1280.
> We use the reported C and M in the original papers.
>
> MatrixLSTM is, in principle, asynchronous. However, since they propose a temporal bin-based representation strategy, and the best results are all achieved with this, we consider it as a synchronous method here.
>
> We did not report GNN results because they are more like a backbone, rather than a representation method. We hope this analysis could help understand the efficiency of different representation methods.
>
> > 3.	In future journal extensions, the authors could consider including more visualization results in the main text to better highlight the advantages of the proposed method.
>
> Thank you for your advice. Due to page limitations, we do not include visualization results in the main context. We plan to add more visualizations in the future version.
>
> [1] Asynchronous Collaborative Graph Representation for Frames and Events, CVPR 2025.

---

> ### Author Response · Authors · 2025-11-26
> **Reply to Questions**
>
> ### Questions:
>
> > 1.	The fitting targets during self-supervised training in this paper are EC and TS. Ideally, after training is completed, the proposed method could become some kind of weighted mixture of EC and TS. So why not directly use EC and TS? For example, in the experiments presented in this paper, what would be the performance change if the input representation were replaced with EC and TS?
>
> Thank you for your question. Since we also use NRP task to predict future events, our method could not be simply a weighted mixture of EC and TS. Results in Ablation Table 5 also demonstrate that adding NRP task could further improve the performance.
>
> Table 1. Comparison with EC and TS on the Gen1.
> | Method | mAP
> | :--- | :---:|
> | EC   | 33.9 |
> | TS   | 49.0 |
> | EVA-L (ours)| 49.7 |
>
> To better support our claim, we compare the performance of directly using EC and TS as event representations on the Gen1 detection task. From the results, directly using EC leads to poor performance (33.9% mAP), while TS could achieve 49.0% mAP. However, our EVA method could further improve the performance to 49.7% mAP, which demonstrates the effectiveness of our method.
>
> Additionally, our method verified that a neural network model can, at least, capture the spatial part of the information delivered by raw events. This could be a starting point to further improve the expressivity of learned representation without imposing additional latency. We think this is also an important contribution of our work.
>
> > 2.	Integration of events and RGB frames is an emerging trend. Could this framework be extended in the future to design a joint asynchronous representation that fuses both modalities? It would be interesting to hear the authors’ thoughts or potential directions on this.
>
> Yes, of course. Since we use a Linear Attention architecture for event processing, it is natural for it to fuse other modalities by means of prefix prompt. Specifically, a straightforward method is to encode RGB images into feature maps by vision backbones, and add the features as "prompt tokens" before the event tokens. Prompt tuning methods could also be investigated to further improve the fusion performance. We will explore this direction in our future work. We are looking forward to your further comments and suggestions on this direction.
>
> > 3.	Introducing NLP-inspired approaches into event representation is a novel idea. What specific advantages does this bring compared to conventional designs? Could the authors provide a more theoretical explanation or justification for this perspective?
>
> Thank you for your question. Our idea of these comes from the difference between frame-based vision and event-based vision. We observed that frame-based vision data are "dense" signals. What we care about is the information within the frame. However, event camera data are "zero redundant" signals, while each event has limited information, what we care about is the relationship between events. This is similar to NLP.
>
> After studying this idea, we find that this NLP-like processing of events could reduce redundant computation. By operating on each event individually, the complexity could be reduced from O(N^2) to O(N) (N is the number of events). By only being concerned about the causal relationship between events via linear attention. Therefore, we think this is a suitable way towards low-latency event-based vision.
>
> > 4.	Looking ahead, could this asynchronous representation technique be incorporated into event-based multimodal large language models [2]? This would be an exciting direction for exploring the intersection between event-based vision and multimodal reasoning.
>
> Thank you for your question. Yes, we think that our EVA model has the potential to be incorporated into event-based multimodal large language models. And we have some preliminary ideas in this direction.
>
> First, EVA could do patch-wise encoding, and the embeddings of each patch could be directly used as the inputs of large language models.
>
> Then, we would like to explore methods to scale up our EVA. Since our model is still small (1M parameters), we would like to explore larger EVA models (100M+ parameters) by scaling up the model size and pretraining on large-scale event datasets. By combining large-scale event-based vision encoders and LLMs, we think it could help to better understand event data.
>
> Finally, we would expect to better leverage the low-latency advantage of our asynchronous encoder in multimodal LLMs. We want to find ways to make the language model also be "event-driven", thus could asynchronously process events. This could significantly reduce the computation cost of LLMs when processing event data.
>
> Since this is an interesting and challenging direction, we would like to explore it in our future work. We are looking forward to your further comments and suggestions on this direction. Thank you!
>
> [2] EventGPT: Event Stream Understanding with Multimodal Large Language Models, CVPR 2025.

---

### Official Review · Reviewer_4uAK · 2025-10-31

**Soundness:** 2
**Presentation:** 2
**Contribution:** 2
**Rating:** 4
**Confidence:** 5

**Summary:**

This paper introduces EVA (EVent Asynchronous feature learning), a novel Asynchronous-to-Synchronous (A2S) framework for learning expressive and generalizable representations from event streams. The key idea is to treat events as “language tokens” and borrow concepts from language modeling, including linear attention and self-supervised feature learning, to construct a unified A2S architecture. EVA demonstrates strong results on multiple benchmarks—DVS128-Gesture and N-Cars for recognition tasks—and becomes the first A2S framework capable of performing detection, achieving 47.7 mAP on the Gen1 dataset.

**Strengths:**

The proposed framework is well-structured and systematically designed. The overall pipeline is coherent, covering event tokenization, temporal-difference encoding, MVHS feature generation, and self-supervised tasks in an end-to-end manner.

The experimental evaluation is comprehensive; the paper validates the generality of EVA across several datasets and tasks, includes meaningful ablation studies, and provides detailed experimental configurations.

The conceptual alignment between event representation and language modeling is novel and promising, opening opportunities for future research on cross-modal modeling between event-based vision and sequence learning.

**Weaknesses:**

In the supplementary material (Fig. 6(c)), it is unclear whether the visualized features correspond to the MVHS outputs. Section E. Visualization lacks a clear description or interpretation of these results.

While performance metrics on N-Cars and Gen1 are reported, latency, throughput, and scalability analyses are missing. Since A2S methods can, in principle, operate at arbitrary event sampling frequencies, it is important to analyze how performance and efficiency vary with sequence length.

There are also minor presentation and clarity issues. The term “#patches” appears in Section 3.1.3 without prior definition and only reappears in the supplementary material. The notation \(P_r\) in Eqs. (4) and (5) is undefined, and Table 2 is never cited in the main text.

**Questions:**

How is the number of events \(N\) in MVHS determined for different datasets, and how does it affect performance and computational cost?

The motivation for selecting Event Count (EC) and Time Surface (TS) as the self-supervised prediction targets should be clarified, as well as whether other representations have been considered.

Although EVA avoids event accumulation, Table 1 shows an additional latency of 14.7 ms. Comparisons with other learnable event representations such as ERGO (“From Chaos Comes Order”) and EventPillars (“Pillar-based Efficient Representations for Event Data”) under identical backbones would better position EVA’s efficiency and expressivity.

---

> ### Author Response · Authors · 2025-11-27
> **Reply to Weaknesses 1**
>
> Dear reviewer 4uAK,
>
> Thank you for your careful and helpful review and comments on our maniscript. Here are our replies.
>
> ### Weaknesses:
>
> > 1.	In the supplementary material (Fig. 6(c)), it is unclear whether the visualized features correspond to the MVHS outputs. Section E. Visualization lacks a clear description or interpretation of these results.
>
> We sincerely thank your critics regarding the visualization results in Fig.6 and Section E (Section F in the new version).
>
> For the visualized features in Fig. 6(c), they are exactly the MVHS layer outputs. To clarify this, we add additional explanations in Fig. 6(c) in the revised manuscript.
>
> We thank you for pointing out the inadequate analysis of the visualization results. For this, we add new description and interpretations of the visualization results in this section.
> Several observations are supplied from the visualization results:
>
> (1) From the similarity between the predicted EC and TS in MRP task (Fig. 6 (b)), we could demonstrate that the EVA model could effectively learn to capture the majority of spatial information from the raw event sequential inputs.
>
> (2) From the analysis of the predicted EC in NRP task (Fig. 6 (a)), we could verify that the EVA model could effectively predict the future, but not simply copy the current input.
>
> (3) We find that several channels of the learned feature show patterns similar to the handcrafted representations, but with different windows of temporal accumulation. These could indicate that the model is learning to capture temporal dynamics in a more flexible manner than fixed handcrafted representations, which is also oberved in other representation works like ERGO, which optimize a multi channel event representation via information maximization [1].
>
> [1] Zubić, Nikola, et al. "From chaos comes order: Ordering event representations for object recognition and detection." Proceedings of the IEEE/CVF International Conference on Computer Vision. 2023.

---

> ### Author Response · Authors · 2025-11-27
> **Reply to Weakness 2**
>
> > 2.	While performance metrics on N-Cars and Gen1 are reported, latency, throughput, and scalability analyses are missing. Since A2S methods can, in principle, operate at arbitrary event sampling frequencies, it is important to analyze how performance and efficiency vary with sequence length.
>
> Thank you for your helpful comments. The latency and throughput analysis of our EVA method is indeed important for evaluating its efficiency. In our original manuscript, we only compare it on DVS128-Gesture becase these metrics of our baseline (ALERT-Tr. +RM) on other datasets are not available. To fully evaluate the latency of our model, we add a detailed analysis of latency and throughput in Appendix I.
>
> For throughput (latency) analysis, we measure the throughput of our EVA model on a NVIDIA 3090 GPU, and the results are shown in Table 1. EVA-L on N-Cars and Gen1 has a slightly lower throughput compared with EVA on DVS128-Gesture, mainly because of its larger model size, which leads to more computation per event (1.2M).
>
> #### Table 1. Throughput of EVA models.
>
> | Model          | EVA  | EVA (d_head=16) | EVA-L  |
> |----------------|-------|------|--------|
> | Parameter (M)        | 0.6  | 0.6  | 1.4   |
> | MACs per event(M)    | 0.6  | 0.61  | 1.2   |
> | Throughput (K/s)     | 611   |  560  | 541  |
> | Latency every 10000 events (ms) | 16.4  | 17.8 | 18.5 |
>
> Additionally, we provide an analysis of the Event Rates of datasets in Table 2. Event Rate refers to the number of events per second, while Patch Event Rate refers to the number of events per second in each patch. These metrics could help to better understand the throughput requirement for real-time processing if we are processing real-world event streams from such datasets. For DVS128-Gesture and N-Cars, the Event Rate are much lower than the throughput of our EVA model. For Gen1, the Event Rate (618K/s) is higher than our EVA-L throughput (541K/s).
> However, considering that we use a Patch-wise Encoding (PWE) strategy: each patch processing events independently, and thus could be computed in parallel (e.g. on multiple devices). Therefore, as long as the hardware could compute at a higher speed than the Patch Event Rate, real-time processing could be, in principle, achieved. The Patch Event Rate of Gen1 is only 2.2K/s which is lower than the throughput of our EVA-L model (541K/s). We believe this could help understand the latency, throughput, and scalability of our EVA method.
>
> #### Table 2. Event Rates of datasets.
>
> | Dataset          | DVS128-Gesture | N-Cars | Gen1  |
> |------------------|----------------|--------|-------|
> | Resolution       | 128x128        | 100x120| 240x304|
> | Avg. Event Rate (K/s) | 58             | 39     | 618   |
> | Patch Size       | 16x16          | 16x16  | 16x16 |
> | Patch Event Rate (K/s)| 0.9   | 4.1 | 2.2 |
>
>
> For the performance and efficiency analysis regarding different sequence length, we find this is indeed an interesting and important problem. Considering that in real-world applications, an event camera could generate events at different rates depending on the scene dynamics, and incase of high event rates which surpass the throughput of the aysnchronous encoder, it is critical to understand how the model performs under such conditions.
>
> To analyze this, we conduct experiments on DVS128Gesture dataset. We downsample the event sequences to lower event rates (2x, 4x, 8x, 16x), and evaluate the recognition accuracy and throughput of our EVA method. We freeze the pretrained weights of EVA encoder as in the original experiments to simulate the possible downsampling scenarios.
> The results are shown in Table 3. For a shorter sequence length, the throughput could be correspondingly increased (1x -> 16x), which could help to meet the real-time processing requirement under high event rates.
> The accuracy will decrease due to the information loss from downsampling (Acc. 96.% -> 93.1%).
> However, with a 8x downsampling, the accuracy (95.1%) is still higher than the baseline ALERT-Tr. +RM method (94.1%), with a higher throughput (4888K/s vs 1412K/s).
> These experiment results are quite interesting for a flexible trade-off between accuracy and efficiency, and we will add these in the main content of our revised manuscript.
>
> #### Table 3. Performance and efficiency under different event rates.
>
> | Sequence length  | Sample Acc. | File Voting Acc. | Throughput (K/s) |
> |---|---|---|---|
> | 8192        |  92.9    |  96.9            |  611 (original)          |
> | 4096        |  91.8    |  96.2            |  1222 (2x)               |
> | 2048        |  91.3    |  95.3            |  2444 (4x)               |
> | 1024        |  90.0    |  95.1            |  4888 (8x)               |
> | 512         |  88.4    |  93.4            |  9776 (16x)              |
> |8192 (ALERT-Tr.+RM)| 84.6 |  94.1            |  1412 (original)       |
> |8192 (ALERT-Tr.+LMM)|72.6 |  86.2            |  2100 (original)       |

---

> ### Author Response · Authors · 2025-11-27
> **Reply to Weakness 3 and Question 1**
>
> ### Weakness
>
> > 3.	There are also minor presentation and clarity issues. The term “#patches” appears in Section 3.1.3 without prior definition and only reappears in the supplementary material. The notation (P_r) in Eqs. (4) and (5) is undefined, and Table 2 is never cited in the main text.
>
> We sincerely thank you for your careful review and pointing out these issues. We will fix three issues  in the revised manuscript.
>
> ### Questions
>
> > 1.	How is the number of events (N) in MVHS determined for different datasets, and how does it affect performance and computational cost?
>
> Thank you for your question. The number of output channels (denoted as N in the manuscript) in MVHS, is determined to have a similar parameter size and feature size with our baseline (ALERT-Tr.) for a fair comparison.
>
> In detail, we keep same hidden size (D=128) and feature size (1024) as ALERT Transformer.
> For MVHS with shape (N, D/N, D/N), the feature size is calculated as N * (D/N) * (D/N). To make the feature size equal to 1024, we have N * (D/N)^2 = 1024, thus N=16 (which we used for DVS128-Gesture and N-Cars).
>
> For Gen1 dataset however, we use RVT as the downstream backbone. Therefore, to be consistent with RVT implementation (which require a Nx240x304 input), we set D/N=P to ensure the output EVA representation has same Height and Width as the original event sensor. Thus, we have N=D/P=128/16=8 for EVA and N=192/16=12 for EVA-L.
> The feature size therefore would be N * (D/N) * (D/N) = N * P * P, which is 2048 for EVA and 3072 for EVA-L， which is larger than our baseline. Therefore, to keep a fair comparison, we use only half of the channels of the EVA representation as the input of RVT backbone. Thus the feature shape is (N/2, D/N, D/N), where N=8 for EVA and N=12 for EVA-L. We hope this could clarify your question.
>
> For the effect of N on performance and computational cost, a larger N would lead to smaller feature size (which is D^2/N). This would reduce computation cost but decrease performance. Here we summerize the results of different N on N-Cars dataset in Table 4. For a smaller N, the feature size is larger, leading to higher accuracy but more MACs and lower throughput. However, this effect is not very significant. We think N could be flexibly selected based on the computation budget.
>
> #### Table 4 . Effect of N on Performance and Computation Cost on N-Cars dataset.
>
> | Model | N | D | Feature shape  |  MACs  |  Throughput  |  Acc.  |
> | :--- | :---: | :---: | :---: | :---: | :---: | :---: |
> | EVA            | 16 | 128 | (16,8,8)   | 0.60 M |  611K/s   | 91.4% |
> | EVA(d_head=16) | 8  | 128 | (8,16,16)  | 0.61 M |  560K/s   | 91.6% |
>
> Please let us know if you have any further questions or concerns. Thank you!

---

> ### Author Response · Authors · 2025-11-27
> **Reply to Questions 2~3**
>
> ### Questions
>
> > 2.	The motivation for selecting Event Count (EC) and Time Surface (TS) as the self-supervised prediction targets should be clarified, as well as whether other representations have been considered.
>
> Thank you for your question.
>
> (1)	The motivation of MRP, where we predict multiple event-based handcraft representations, comes from the fact that different representation contain different aspect of the information in raw events, and predicting them, like multi-task learning, could force the model to learn a more comprehensive representation. Therefore, by doing this, we hope the model to preserve more information in the raw events.
>
> (2)	Yes, the MRP task could definetly be expanded to other handcrafted representations, and even learned representations, as long as they could be calculated from the raw events. In the experiments, we use EC and TS as SSL targets, mainly comes from 2 considerarions:
>
> - EC and TS are most commonly used representations, and is broadly used in various event-based tasks (recognition, detection, optical flow, segmentation, depth estimation, etc.). Using EC and TS for targets would not decrease the performance.
> - They are easy and fast to compute compared with complex representations. Since our method requires handcrafted representations as targets, we have to compute them from raw events at training. Using other complex representations would extremely decrease the training speed.
>
> (3)	In our current stage of experiments, we tried EST as further representations in the object recognition experiments. However, this will introduce a 2x more pretraining time, and the accuracy improvement is marginal (<0.5% on DVS128Gesture). We suspect that, since our EVA representation (learned w/o EST), could already surpass EST in most experiments (N-Cars, N-Caltech101), adding additional supervision on these datasets would not benefit too much. For Gen1, since the dataset is large, from the perspective of efficiency, we do not consider adding other representations in our experiment.
>
> (4)	Future researches. Here we explore future possible improve directions for the SSL task. Since EVA is good at predicting the future (by NRP), we could explore combining it with JEPA[2]. We hope this could help the model to learn more beyond handcrafted representations, and further improve its expressivity.
>
> [2] Assran, Mahmoud, et al. "Self-supervised learning from images with a joint-embedding predictive architecture." Proceedings of the IEEE/CVF Conference on Computer Vision and Pattern Recognition. 2023.
>
> > 3.	Although EVA avoids event accumulation, Table 1 shows an additional latency of 14.7 ms. Comparisons with other learnable event representations such as ERGO (“From Chaos Comes Order”) and EventPillars (“Pillar-based Efficient Representations for Event Data”) under identical backbones would better position EVA’s efficiency and expressivity.
>
> Thank you for your valuable suggestion. We compare our EVA method with ERGO, EventPillars, and other representing methods with identical Swin Transformer V2 under same experiment settings. The results are shown in Table 5.
>
> #### Table 5. Comparison with other representing methods on Gen1.
> | Method | mAP | Representing latency (ms) |
> | :--- | :---: | :---: |
> | Voxel (bin=12) | 39.5 | 16.3 |
> | ERGO-12 (w aug.)  | 50.4 | > 10 |
> | ERGO-12 (w/o aug.)  | 49.3 | > 10 |
> | Pillar| 53.1 | 11.4 |
> | HOTS | 49.0 | 1.0 |
> | TORE | 43.6 | < 0.1 |
> | EVA-L (ours)| 49.7 | < 0.1 |
>
> Here we want to clarify that, the 14.7 ms latency in Table 1 is the accumulated latency for processing 8192 events. For each event, it takes only 2 $\mu s$ to process, which is much lower that Pillar and ERGO.
> The difference here is that, Pillar and ERGO are synchronous methods, they could not update the events event-by-event, and require waiting for a fixed processing time (10+ ms). Our method, instead, could update event representation in an increamental manner. There fore, we do not need to recompute the entire representation from all the events when a new event arrives, which significantly reduce the representing latency.
>
> From Table 5, our EVA method outperforms all asynchronous methods, and have competitive results compared with dense synchronous representations.
> Specifically, our EVA-L achieves 49.7% mAP, surpassing even some synchronous methods (ERGO-12 w/o aug. 49.3%), while maintaining extremely low representing latency.
> We believe this coud demonstrates the expressivity and efficiency of our EVA framework.
>
> Please let us know if you have any other concerns or questions. Thank you!

---

### Official Review · Reviewer_q375 · 2025-11-04

**Soundness:** 3
**Presentation:** 3
**Contribution:** 2
**Rating:** 6
**Confidence:** 3

**Summary:**

This paper introduces EVA, an asynchronous architecture for event camera data. The key idea is to treat streams of visual events as analogous to language tokens, each carrying incremental information about a scene and to employ linear attention architectures for scalability. The encoder maintains a hidden state that aggregates spatial and temporal structure while supporting fast processing.

**Strengths:**

* Tackles a timely challenge as it is yet unclear how to process event camera inputs
* Interesting ideas to use a token based approach with a linear transformer
* The self-supervised aspect with the multi-representation training is seemingly novel and interesting

**Weaknesses:**

Overall, the major concerns I have with this work are that the results are not very compelling.

The authors only evaluate two task (object detection and recognition). Given the authors are proposing a new architecture, it would be good to have at least another slightly different task to demonstrate the robustness of their architecture is not clear.

The proposed approach achieves does not achieve the best accuracy or latency, but is another tradeoff point in the space.

If the key benefits are the fast processing, then it would be good to have a more thorough analysis of where the efficiency benefits come from and whether real-time performance can actually be achieve with their approach on realistic GPUs for edge computing.

The effectiveness of the approach with self-supervised learning using handcrafted representations is not convincing as the method to have it learn the right representations.

**Questions:**

Why is your latency higher when more parameters are used in Table 1?

---

> ### Author Response · Authors · 2025-11-26
> **Reply to Weakness 1**
>
> Dear reviewer q375,
>
> Thank you for your helpful and careful review. Here are our replies.
>
> ###  Weaknesses:
> > 1.	Overall, the major concerns I have with this work are that the results are not very compelling.
> The authors only evaluate two task (object detection and recognition). Given the authors are proposing a new architecture, it would be good to have at least another slightly different task to demonstrate the robustness of their architecture is not clear.
>
> Thank you for your helpful comments. To better demonstrate the robustness of the proposed architecture, we follow your suggestions and validate our method on the optical flow task. However, due to computational resource limitations, the results are not available yet. We would provide the results immediately after it finishes.
>
> Meanwhile, we provide extended experiments for recognition and detection tasks. For recognition, we add results on the N-Caltech101 dataset created by capturing moving static datasets. For detection, we provide further comparison of our EVA feature with other event-representing methods under an identical backbone (Swin v2). The results are shown in Tables 1 & 2.
>
> From the Tables 1 & 2, our method outperforms all asynchronous methods, and has competitive results compared with dense synchronous representations.
> Specifically, in Table 1, our EVA-L achieves 86.3% accuracy, which is only 0.3% lower than the best synchronous method E2VID (86.6%), and outperforms all the asynchronous methods by a large margin (at least 10%).
> In Table 2, our EVA, with a Swin v2 backbone, achieves 49.7% mAP, while maintaining extremely low representing latency.
> This further validates the effectiveness of our EVA framework.
>
> #### Table 1. Recognition results on N-Caltech101 dataset. A. denotes asynchronous methods, S. denotes synchronous methods.
> | Model | Type | Acc. (%) |
> |-------|------|------|
> | HOTS      | A. | 21.0 |
> | NVS-S     | A. | 67.0 |
> | AEGNN     | A. | 66.8 |
> | Asynet    | A. | 74.5 |
> | FARSE-CNN | A. | 68.7 |
> | EST       | S. | 81.7 |
> | Matrix-LSTM      |  S.  | 84.3 |
> | Pillar           |  S.  | 85.3 |
> | E2VID            |  S.  | **86.6** |
> | EVA-L (ours)     |  A2S | $\underline{86.3}$ |
>
>
> #### Table 2. Detection results on the Gen1 dataset with an identical backbone.
>
> | Method | mAP | Representing latency (ms) |
> | :--- | :---: | :---: |
> | Voxel (bin=12) | 39.5 | 16.3 |
> | ERGO-12 (w aug.)  | 50.4 | > 10 |
> | ERGO-12 (w/o aug.)  | 49.3 | > 10 |
> | Pillar| 53.1 | 11.4 |
> | HOTS | 49.0 | 1.0 |
> | TORE | 43.6 | < 0.1 |
> | EVA-L (ours)| 49.7 | < 0.1 |

---

> > ### Author Response · Authors · 2025-12-03
> > **Reply to Weakness 1**
> >
> > > 1. Overall, the major concerns I have with this work are that the results are not very compelling. The authors only evaluate two task (object detection and recognition). Given the authors are proposing a new architecture, it would be good to have at least another slightly different task to demonstrate the robustness of their architecture is not clear.
> >
> > Here we also evaluate our method on optical flow estimation. We on the MVSEC dataset and follow the experimental settings of EV-FlowNet. We compare average endpoint error (AEE). As shown in Table 5, our method is also applicable to optical flow estimation.  Our method outperforms the asynchronous method HUGNet-2, while giving similar results to the synchronous method EV-FlowNet but without representing latency, which further demonstrates the effectiveness of our method.
> >
> > #### Table 5: Optical flow estimation results on the MVSEC dataset.
> >
> > | Method                                   | Type | AEE |
> > |------------------------------------------|------|-----|
> > | HUGNet-2 (Dampfhoffer et al., 2025)      | A.   | 1.52 |
> > | EV-FlowNet (Zhu et al., 2018)            | S.   | 0.96 |
> > | Spike-FlowNet (Lee et al., 2020)         | S.   | **0.79** |
> > | EVA-L (ours)                             | A2S  | 1.01 |

---

> ### Author Response · Authors · 2025-11-26
> **Reply to Weakness 2**
>
> ### Weakness:
> > 2.	The proposed approach achieves does not achieve the best accuracy or latency, but is another tradeoff point in the space.
> If the key benefits are the fast processing, then it would be good to have a more thorough analysis of where the efficiency benefits come from and whether real-time performance can actually be achieve with their approach on realistic GPUs for edge computing.
>
> Thank you for your helpful comments. Indeed, our EVA method does not achieve the best accuracy when compared with SOTA synchronous methods (ERGO, Pillar). Similarly, our method does not achieve the lowest latency when compared with some simple representations (ALERT-Tr.).
>
> (1) From the latency perspective, it is not necessary to really reduce the latency to the lowest. If we could process events faster than their arrival rate, real-time processing could be achieved. Additionally, since we use a Patch-wise Encoding (PWE) strategy, each patch processes events independently, and thus could be computed in parallel. Therefore, as long as the hardware could calculate events at a higher speed than the Patch Event Rate, real-time processing could, in principle, be achieved.
>
> Here we provide an analysis of the (Patch) Event Rates of different datasets in Table 3, and a throughput result of our EVA method on a common GPU (NVIDIA RTX 3090) in Table 4.
>
> #### Table 3. Event Rates of different datasets. (Event Rate: number of events per second; Patch Event Rate: number of events per second in each patch)
>
> | Dataset          | DVS128-Gesture | N-Cars | Gen1  |
> |------------------|----------------|--------|-------|
> | Resolution       | 128x128        | 100x120| 240x304|
> | Avg. Event Rate (K/s) | 58             | 39     | 618   |
> | Patch Size       | 16x16          | 16x16  | 16x16 |
> | Patch Event Rate (K/s)| 0.9   | 4.1 | 2.2 |
>
> #### Table 4. Throughput of the EVA method on NVIDIA RTX 3090 GPU.
>
> | Model          | EVA  | EVA-L  |
> |----------------|-------|--------|
> | Parameter (M)        | 0.6  | 1.4   |
> | MACs per event(M)    | 0.6  | 1.2   |
> | Throughput (K/s) | 611   | 541  |
>
> From Tables 3 & 4, we could see that our EVA method could process events at more than 500K events/s on a common GPU, which is much higher than the Patch Event Rate of all the datasets (max 4.1K events/s). These demonstrate that although our method does not achieve the lowest latency, it could theoretically achieve the desired real-time processing.
>
> (2) From the accuracy perspective, our EVA method reduces the representation latency by 10+ ms compared with the SOTA synchronous methods (ERGO, Pillar), while only sacrificing less than 3% accuracy. Consider that the inference latency of the backbone is around 10 s of ms, the extra 10+ ms latency introduced by synchronous representing methods is not negligible.
> We think this is a reasonable trade-off for real-time applications, where low latency is critical.
>
> (3) Additionally, to further verify the implementation capability of our EVA method on edge devices. We implement a C++ version of our EVA model, and achieve a throughput of 8K events/s on a single thread of CPU (Intel Xeon Gold 6130 CPU), which is higher than the Patch Event Rate of event camera datasets.
> Since the EVA model is highly parallelizable without complicated logic, it could also be deployed on AI accelerators (like FPGA, VPU, etc.) for further speedup. However, since deployment on edge devices is not the main focus of this work, we leave it as future work.
>
> Since the latency-accuracy trade-off is a common interest problem regarding the EVA method, we add a discussion in Appendix I. We are looking forward to your further comments and insight about this problem. Thank you!

---

> ### Author Response · Authors · 2025-11-26
> **Reply to Weakness 3 and Questions**
>
> ### Weakness
> > 3. The effectiveness of the approach with self-supervised learning using handcrafted representations is not convincing as the method to have it learn the right representations.
>
> We sincerely thank your comments. From the perspective of designing the EVA model, we use handcrafted representations (EC, TS) for 2 main considerations.
>
> (1) The handcrafted representations are widely used in event-based vision tasks and have been proven to be effective in capturing the majority of the spatial information of event data. Our model is expected to learn a general representation of event data by leveraging these well-established representations as a foundation.
>
> (2) Different handcrafted representations capture different aspects of event data. By using multiple handcrafted representations (EC and TS), our model can learn a more comprehensive representation that combines the strengths of each representation.
>
> Additionally, the new results in Table 2 could further validate the effectiveness of learning from handcrafted representations. Under the same experimental settings (w/o aug.), our EVA even outperforms ERGO-12, which is a theoretical optimal representation from the perspective of information theory.
> We believe this demonstrates the capability of our model to learn effective representations from handcrafted targets.
>
> ### Questions:
> > 1.	Why is your latency higher when more parameters are used in Table 1?
>
> Thank you for your question. In Table 1, EVA (0.62M) has 14.7 ms latency, while ALERT-Tr.(+RM) (1.41M) has only 5.8 ms latency.
> The reason why our method has higher latency is that we use a more complex event-by-event updating module (RWKV-6), while ALERT-Tr.(+RM) uses a simple MaxPooling layer for event-by-event updating, which is better supported by the current deep learning frameworks.
>
> What we want to emphasize is that the latency provided here is the accumulated latency of the entire sequence with 8192 events. For each event, it takes only 2 $\mu s$ to process. Consider that on the DVS128Gesture dataset, the average Event Rate is about $6\times 10^4$ events/s, and the Patch Event Rate is only $1\times 10^3$ events/s. Our method meets this throughput requirement.

---

### Comment · Area_Chair_RBzk · 2025-11-27

Dear Authors,

I notice you did not respond to  Reviewer 4uAK yet.

---

> ### Author Response · Authors · 2025-11-27
>
> Dear AC,
>
> Thank you for your kind reminder. We have updated our responses to Reviewer 4uAK.

---

### Author Response · Authors · 2025-12-03
**Summary of the discussion**

Dear Area Chair,

Thank you for your time on our manuscript. We are very happy about the quality of reviews we received for this paper. We actually agree with most of the feedback, and we have made progress in addressing all of the main reviewers' concerns. Here we would like to summarize the discussion for your reference.

In this submission, we propose EVA, an event-based asynchronous-to-synchronous (A2S) framework that asynchronously encodes raw events into tensor-based representations, diminishing the representing latency of synchronous representations, and benefiting downstream tensor-based neural networks as well. Our EVA model, inspired by the language-event analogy, leverages a linear attention for asynchronous event embedding, maintaining high efficiency and expressivity. Basically, our method outperforms asynchronous methods, and achieves competitive results compared with synchronous methods, while having no representing latency.

The main concerns raised by the reviewers and our corresponding responses are as follows:

1) **Detailed latency analysis.** In the original submission, we compare the latency of our EVA model with the baseline (ALERT-Tr.) in Section 4.1 and show our better performance. The reviewers also asked for a more detailed latency(throughput) analysis. The main questions are about (i) what the throughput of all our EVA and EVA-L models is, and could it can achieve real-time performance, (ii) the low-latency advantage compared with other synchronous representations, and (iii) how the latency varies with sequence length and event camera resolution.

For these concerns, we provided additional experiments and analysis in the rebuttal. (i) We provide the throughput of our EVA and EVA-L models, and we also provide the statistics of the event camera datasets to evaluate the real-time performance. Our EVA model has a larger throughput than the patch event rates, thus achieving real-time performance. (ii) We compare the latency of our EVA model with SOTA synchronous representations (e.g., ERGO, Pillar) and show that our model has a low-latency advantage of more than 100x speedup. We present a theoretical complexity analysis of our model and synchronous representations for better understanding. (iii) We present results on how the latency varies with sequence length and event camera resolution. Besides, we also evaluate the model under different downsampling rates to show the trade-off between computation cost and performance. By doing so, we provide a comprehensive understanding of the latency and throughput of our EVA model, and we believe this addresses the reviewers' concerns.

We add the latency and throughput analysis in Appendix I. Since this part is commonly of interest, we plan to move this part to the main paper if the paper could be accepted and we have more pages to use.

2) **Extended experiments.** In the original submission, we give a comprehensive comparison with our baseline (ALERT-Tr.). We also evaluate our model on detection tasks, which were not achieved by previous A2S work. The reviewers asked for more experiments on (i) more challenging classification datasets, (ii)  other tasks such as optical flow estimation, (iii) comparison with other synchronous representations under identical backbones, and (iv) a high-resolution (1280x720) dataset.

For these concerns, we provided additional experiments.
(i-iii) We evaluate our model on additional datasets N-Caltech101 dataset and the MVSEC dataset. We also evaluate it on Gen1 with an identical Swin v2 backbone. As expected, our model gives better results compared with the asynchronous method. For synchronous methods, we have competitive results but do not introduce representing latency, thus have a low-latency advantage.  The results are presented in Appendix A.
(iv) We explain that high-resolution datasets are a common open problem for asynchronous event camera processing, and give a theoretical analysis in Appendix I.
We believe these additional experiments address the reviewers' concerns.

3) **Interpretation of visualization**. The reviewers asked for a more detailed interpretation of the visualization in Figure 6, Appendix F. We provide a more detailed explanation of the visualization in the rebuttal.

4) **Clarity issues**. We also address several clarity issues raised by the reviewers in the rebuttal, including citation, figures, and writing issues. We believe these improvements enhance the overall quality of the paper.

5) **Future works**. We also answer the reviewers' questions regarding potential future works, including RGB-event fusion and combining with MLLMs. We believe these discussions provide valuable insights for future research directions.

We believe that we have addressed all the main concerns raised by the reviewers. And we hope that our summary helps you in your decision-making process.

Thanks for your consideration.

---

### Meta-Review · Area_Chair_jECY · 2026-01-07

**Summary:**

Reviewers found the proposed EVA framework novel and well motivated, with a compelling language-inspired perspective for asynchronous event processing. However, they raised concerns about whether the original experimental evidence was sufficient to support a new architectural framework, noting limited task diversity and requesting stronger validation of generality.

A central issue was the lack of detailed latency, throughput, and scalability analysis, as well as clearer comparisons with strong synchronous representations under identical backbones. Additional concerns included the justification of the self-supervised targets based on handcrafted representations and several clarity and presentation issues. These points led to mixed reviewer recommendations around the acceptance threshold.

The AC checked the paper and agreed with the reviewers for the major comments.

Considering this paper's novelty and well-positioned motivation, the AC recommends an Accept for ICLR community. The authors are suggested to incorporate all comments in the camera-ready revision.

**Reviewer Concerns:**

- Some concerns remain partially outstanding. While the rebuttal includes theoretical arguments and limited analysis regarding high-resolution event streams, reviewers’ requests for empirical validation on truly high-resolution datasets could not be fully satisfied due to computational constraints.

- Additionally, although efficiency comparisons were broadened, direct end-to-end runtime evaluations across a wider range of open-source representations and hardware settings remain limited. These remaining issues are primarily empirical in nature and do not undermine the core technical contributions, but they leave some open questions regarding large-scale deployment.

**Reviewer Scores:**

There was no discussion from all four reviewers. If they were to participate fully, there might be change of scores.

Reviewer q375: Likely to increase slightly (e.g., from 6 to 7).

Reviewer 4uAK: Likely to move from marginally negative to neutral or slightly positive (e.g., from 4 to 5–6).

Reviewer n32k: Likely to remain stable or increase slightly (e.g., from 6 to 6–7).

Reviewer A3qJ: Likely to remain unchanged or increase marginally (e.g., from 6 to 6–7).

---

### Decision · Program_Chairs · 2026-01-26

Accept (Poster)